# Facilitating Graph Neural Networks with Random Walk on Simplicial Complexes

**Cai Zhou**
Tsinghua University
zhouc20@mails.tsinghua.edu.cn

**Xiyuan Wang**
Peking University
wangxiyuan@pku.edu.cn

**Muhan Zhang**
Peking University
muhan@pku.edu.cn

## Abstract

Node-level random walk has been widely used to improve Graph Neural Networks. However, there is limited attention to random walk on edges and, more generally, on $k$-simplices. This paper systematically analyzes how random walk on different orders of simplicial complexes (SC) facilitates GNNs in their theoretical expressivity. First, on $0$-simplices or node level, we establish a connection between existing positional encoding (PE) and structure encoding (SE) methods through the bridge of random walk. Second, on $1$-simplices or edge level, we bridge edge-level random walk and Hodge 1-Laplacians and design corresponding edge PE respectively. In the spatial domain, we directly make use of edge level random walk to construct EdgeRWSE. Based on the spectral analysis of Hodge 1-Laplcians, we propose Hodge1Lap, a permutation equivariant and expressive edge-level positional encoding. Third, we generalize our theory to random walk on higher-order simplices and propose the general principle to design PE on simplices based on random walk and Hodge Laplacians. Inter-level random walk is also introduced to unify a wide range of simplicial networks. Extensive experiments verify the effectiveness of our random walk-based methods.

## 1 Introduction

Graph neural networks (GNNs) have recently achieved great success in tasks with graph-structured data, benefiting many theoretical application areas, including combinatorial optimization, bio-informatics, social-network analysis, etc. [11, 29, 16]. Two important aspects to evaluate GNN models are their theoretical expressivity in distinguishing non-isomorphic graphs, and their performance on real-world tasks. Positional encoding (PE) and structure encoding (SE) are widely adopted methods to enhance both theoretical expressivity and real-world performance of GNNs. Generally, PE encodes the information of the nodes' local or global positions, while SE provides information about local or global structures in the graph. For example, Kreuzer et al. [30] uses eigenvectors of the graph Laplacian, Dwivedi et al. [18] proposes to use diagonal elements of the $t$-step random walk matrix, and Bouritsas et al. [9] manually count some predefined structures. There are also some methods based on pair-wise node distances, such as the shortest path distance [31], the heat kernel [20], and the graph geodesic [35]. Although some work theoretically analyzes some of these methods [51], there are still some left-out methods, and people lack a unified perspective to view all these PE and SE designs. Moreover, most existing methods focus only on node data, while PE and SE on edge data as well as some higher-order topological structures are waited to be studied.

In addition to PE and SE, geometric deep learning has recently become a central topic. Researchers are inspired by concepts of differential geometry and algebraic topology, which resulted in many works on simplices and simplicial complexes [8, 7, 47]. Despite their capability to deal with higher-order structures, these simplicial networks should follow orientation symmetry, which brings difficulties in their applications in undirected graphs. This work connects these two separate areas via a central

37th Conference on Neural Information Processing Systems (NeurIPS 2023).

concept: random walk on simplicial complexes. On the one hand, by introducing concepts of higher-order simplicial complexes, we can design more PE and SE methods that are both theoretically and practically powerful. On the other hand, PE and SE greatly facilitate simplicial data and benefit graph learning.

In summary, we first connect a number of existing PE and SE methods through the bridge of node-level random walk on 0-simplices. Then, for 1-simplices or edges, we design two novel sign and basis invariant edge-level PE and SE, namely EdgeRWSE and Hodge1Lap. EdgeRWSE uses an edge-level random walk directly to capture structure information, while Hodge1Lap is based on spectral analysis of Hodge 1 Laplacian, which is closely related to random walk on edges. We further generalize our theory to random walk on higher-order and inter-order simplices to facilitate graph and simplicial learning. Our methods achieve State-Of-The-Art or highly competitive performance on several datasets and benchmarks. Code is available at https://github.com/zhouc20/HodgeRandomWalk.

## 2 Related work

**Theoretical expressivity and Weisfeiler-Lehman test.** Weisfeiler-Lehman tests are a classical family of algorithms to distinguish non-isomorphic graphs. Previous work has built connections between the expressivity of GNNs and the WL hierarchy. Some classical conclusions include that for $k \geq 2$, $k + 1$-dimensional WL is more powerful than $k$-WL. [46] proves that traditional message-passing neural networks (MPNN) are not more powerful than 1-WL. There is another variation of the WL test called the Folklore Weisfeiler-Lehman (FWL) test, and $k$-FWL is equivalent to $k$-WL in expressivity for $k \geq 1$.

**Symmetry in graph and simplicial learning.** Symmetry is a central topic in graph and simplicial learning. In graph learning, node features and edge features need to be permutation (i.e., relabeling of nodes or edges) equivariant, while the graph features should be permutation invariant. In simplicial learning, one needs to further orientation symmetry [47] in an oriented simplicial complex (SC). The incidence relations and the simplicial adjacencies in an oriented SC are altered when the orientations are reversed. The $k$-form remains invariant to this transformation, while the features of $k$-simplices are equivariant in terms of the basis. [32] also state the standard that the graph-level functions (and in the context of SC, $k$-forms) should be invariant to both sign and basis (either of orientation or of space), which is a basic rule for our PE and SE designs.

## 3 Preliminary

**Graphs.** We denote a graph as $G(V, E, A)$, where $V, E$ is the set of nodes and the set of edges, respectively, and $A$ is the adjacency matrix for the nodes. For convenience, we use $n = |V|$ and $m = |E|$ to represent the number of nodes and edges in the graph $G(V, E, A)$. In an undirected graph, for any $u, v \in V$, we have $(u, v) \in E \Leftrightarrow (v, u) \in E$. Let $\mathcal{N}(v, G) = \{u \in V | (u, v) \in E\}$ denote the set of neighbors of node $v$ in graph $G$. Let diagonal matrix $D = diag(d_1, ..., d_n)$, where $d_i$ is the degree of node $v_i$.

The transition matrix of a typical random walk at node level is $P = D^{-1}A$, which indicates that in each step the walk moves from the current node $v$ to one of its neighboring nodes $u \in \mathcal{N}(v, G)$ with equal probabilities. Consequently, a $t$ step of the aforementioned random walk corresponds to a transition matrix $P^t$.

**Discrete Hodge Laplacian of abstract simplicial complex.** An abstract simplicial complex $\mathcal{K}$ on a finite set $V$ is a collection of subsets of $V$ that is closed under inclusion. In our paper, $V$ will be a vertex set $[n] = \{1, 2, ..., n\}$ if without special statement. An element of cardinality $k + 1$ is called a $k$-face or $k$-simplex of $\mathcal{K}$. For instance, 0-faces are usually called vertices, 1-faces are directed edges, and 2-faces are 3-cliques (triangles) with an orientation. We denote the collection of all $k$-faces of $\mathcal{K}$ as $S_k(\mathcal{K})$. The dimension of a $k$-face is $k$, and the dimension of a complex $\mathcal{K}$ is defined as the maximum dimension of the faces in $\mathcal{K}$.

The definition of neighbors of simplices is crucial in this paper. Two $k + 1$-simplices sharing a collective $k$-face are called $k$-down neighbors, and two $k$-simplices sharing a collective $k + 1$-simplex are called $k + 1$-up neighbors. Generally, a face $F$ is chosen as an ordering on its vertices and is said

to be oriented, denoted by $[F]$. For any permutation element $\sigma \in \mathcal{G}_{k+1}$ where $\mathcal{G}_{k+1}$ is the symmetric group of permutations on $\{0, ..., k\}$, two orders of vertices transformed by $\sigma$ are said to determine the same orientation if $\sigma$ is an even permutation and opposite if $\sigma$ is odd.

In the Hilbert space, the matrix representations of boundary and coboundary operators are adjacency matrices of order $k$ and $k+1$ simplices. In order to keep coordinate with most existing literature, we write the adjacent matrix of $k$-th and $k+1$-th simplices as $\mathbf{B}_{k+1} \in \mathbb{R}^{|S_k| \times |S_{k+1}|}$. $\mathbf{B}_{k+1}[i, j] = 1$ if the $i$-th $k$-simplex and $j$-th $k + 1$-simplex are adjacent and share the same direction, $\mathbf{B}_{k+1}[i, j] = -1$ if adjacent with opposite directions, and 0 if they are not adjacent. For example, $\mathbf{B}_1$ is the node-to-edge incidence matrix.

In discrete Hodge-deRham theory, the $k$-th order Hodge Laplacian is defined as

$$\mathbf{L}_k = \mathbf{B}_k^* \mathbf{B}_k + \mathbf{B}_{k+1} \mathbf{B}_{k+1}^* \tag{1}$$

where $\mathbf{B}_k^* = \mathbf{B}_k^T$ is the adjoint of $\mathbf{B}_k$ and is equivalent to the transpose of $\mathbf{B}_k$ in Hilbert space. A special case is that when $k = 0$, $\mathbf{B}_0$ is not defined and $\mathbf{L}_0 = \mathbf{B}_1 \mathbf{B}_1^* = \mathbf{D} - \mathbf{A}$ is exactly the graph Laplacian. We refer readers to Appendix C.2.2 for an illustrative calculation example of Hodge Laplacians. In our following texts, we will make use of higher-order Hodge Laplacians such as $\mathbf{L}_1$ rather than previously used $\mathbf{L}_0$ alone.

The kernel space of $\mathbf{L}_k$ is called the $k$-th cohomology group: $\tilde{\mathcal{H}}^k(\mathcal{K}, \mathbb{R}) := \ker(\mathbf{B}_{k+1}^*)/\mathrm{im}(\mathbf{B}_k^*) \cong \ker(\mathbf{B}_{k+1}^*) \cap \ker(\mathbf{B}_k) = \ker(\mathbf{L}_k)$. We will write $\tilde{\mathcal{H}}^k(\mathcal{K}, \mathbb{R})$ simply as $\tilde{\mathcal{H}}^k$ without causing confusion. The kernel spaces of Hodge Laplacians are closely associated with harmonic functions and will play an important role in our following analysis. Particularly, the multiplicity of zero eigenvalues of $\mathbf{L}_k$, or the dimension of null space of Hodge $k$-Laplacian $\ker(\mathbf{L}_k)$, is called the $k$-th Betti number $\beta_k$ [23]. This is exactly the number of cycles composed of $k$-simplicials that are not induced by a $k$-boundary, or intuitively, $k$-dimensional "holes" in the simplicial complex $\mathcal{K}$. For example, zero eigenvalues and their eigenvectors of $\mathbf{L}_0$ are associated with the 0-th cohomology group of the graph, corresponding to the connected components of the graph. The zero eigenvalues and eigenvectors of $\mathbf{L}_1$ are associated with cycles (in the usual sense), and those of $\mathbf{L}_2$ correspond to cavities. We refer readers to Appendix C.2.2 for detailed explanations and illustrative examples of cohomology groups.

## 4  Random walk on 0-simplices

Random walk on 0-simplices or at node level has been studied systematically. Previous work has established comprehensive analysis on the theoretical properties of node-level random walk, which provide theoretical insights into the design of random walk-based methods. However, there is still limited research on the theoretical expressivity of random walk-based positional encoding (PE) and structure encoding (SE) methods. In this section, we establish connections between several PE and SE with node-level random walk, and provide theoretical expressive power bounds for them.

**RWSE.** [52] and Dwivedi et al. [18] propose a structure encoding method based on node-level random walk, which we denote as RWSE. Concretely, RWSE considers $K$ steps of random walk at the node level of the graph, obtaining $\mathbf{P}, \mathbf{P}^2, ..., \mathbf{P}^K$. Then the method only takes into account each node's return probabilities to itself, i.e. the diagonal elements of $\mathbf{P}^k$, $k = 1, 2, ..., K$. For each node $v_i$, the RWSE feature is $h_i^{RWSE} = [\mathbf{P}_{ii}, \mathbf{P}_{ii}^2, ..., \mathbf{P}_{ii}^K]$. Compared with encoding methods based on graph Laplacian eigenvalues and eigenvectors, this method is sign and basis invariant. It internally captures some structure information within $K$-hops and achieves impressive results in experiments [38]. However, there are limited investigations on the theoretical expressivity of RWSE and its extensions. Here, we provide a theoretical bound of positional and structure encoding methods based on random walk transition matrix $\mathbf{P}$.

**Theorem 4.1.** *RWSE is strictly less powerful than* 2*-FWL, i.e. RWSE $\prec$ 2-FWL.*

The above expressivity bound holds because 2-FWL can simulate the multiplication and injective transformations of a matrix, including the adjacency matrix $\mathbf{A}$. Therefore, 2-FWL is capable of obtaining $\mathbf{P}^k, k \in \mathbb{N}$. Specifically, a block of PPGN [34] can simulate one time of matrix multiplication. Moreover, RWSE is strictly less expressive than 2-FWL, since it loses much structure information when taking the diagonal elements of $\mathbf{P}^k$ only. In other words, RWSE is a summary of full random walk transition probabilities (on spatial domain), which accelerates calculation at the cost of losing expressivity.

**Resistance distance and random walk.** In addition to RWSE, there are a number of positional encoding methods closely related to the node-level random walk. A.K. et al. [2], Zhang et al. [51] connect commute time in random walks with resistance in electrical networks, which can be used as a PE method called resistance distance (RD). Zhang et al. [51] prove that RD and shortest path distance (SPD) [31] are both upper-bounded by 2-FWL in expressive power.

**Positive definite kernels based on graph Laplacian spectrum.** Graph Laplacian, or Hodge 0-Laplacian as we refer to later, is closely connected with random walk on graph. The definition of graph Laplacian is $\mathbf{L}_0 = \mathbf{D} - \mathbf{A} = \delta_0^* \delta_0 = \Delta_0$. Through the spectrum of $\mathbf{L}_0$, we are able to define a family of positive definite kernels on graphs [42] by applying a regularization function $r$ to the spectrum of $\mathcal{L}_0$: $K_r = \sum_{i=1}^m r(\lambda_i) \mathbf{u}_i \mathbf{u}_i^T$, where $\mathbf{L}_0 = \sum_i \lambda_i \mathbf{u}_i \mathbf{u}_i^T$ is the eigenvalue decomposition. For example, the heat kernel or the diffusion kernel [20] can be incorporated if $r(\lambda_i) = e^{-\beta \lambda_i}$. Other methods directly use eigenvectors as PE [30]. These results imply that spectral analysis of graph Laplacians can also inspire more powerful PE and SE, and we will generalize graph Laplacian $\mathbf{L}_0$ to arbitrary order of Hodge $k$ Laplacians in the following section to facilitate graph learning.

# 5 Random walk on 1-simplices

While node-level random walk has been widely studied, edge-level random walk is still limited. In this section, we will first introduce Hodge 1 Laplacian $\mathbf{L}_1$, as well as its connection with random walk on 1-simplices (in the lifted space) and thus edges of undirected graph. Analogous to node-level RWSE, we introduce EdgeRWSE, a more theoretically powerful PE for edges. Furthermore, we systematically analyze the spectra of $\mathbf{L}_1$ and propose a novel Hodge1Lap PE, the first sign and basis invariant edge-level positional encoding that make use of the spectra of $\mathbf{L}_1$ instead of the previously adopted $\mathbf{L}_0$ only.

## 5.1 Normalized Hodge-1 Laplacian and edge-level random walk

**Theoretical analysis of edge-level random walk.** The standard Hodge $k$-Laplacian is $\mathbf{L}_k = \mathbf{B}_k^* \mathbf{B}_k + \mathbf{B}_{k+1} \mathbf{B}_{k+1}^*$, and there are a number of normalized Hodge Laplacian because the normalization is rather flexible. Schaub et al. [41] propose a normalized form for Hodge 1-Laplacian $\mathbf{L}_1$ with a clear interpretation of a random walk in the lifted edge space. Concretely,

$$\tilde{\mathbf{L}_1} = \mathbf{D}_2 \mathbf{B}_1^* \mathbf{D}_1^{-1} \mathbf{B}_1 + \mathbf{B}_2 \mathbf{D}_3 \mathbf{B}_2^* \mathbf{D}_2^{-1} \tag{2}$$

where $\mathbf{D}_2$ is the diagonal matrix with adjusted degrees of each edge $\mathbf{D}_2 = \max(diag(|\mathbf{B}_2|\mathbf{1}), I)$, $\mathbf{D}_1$ is the diagonal matrix of weighted degree of nodes $\mathbf{D}_1 = 2 \cdot diag(|\mathbf{B}_1|\mathbf{D}_2\mathbf{1})$, and $\mathbf{D}_3 = \frac{1}{3}\mathbf{I}$.

To interpret this normalized Hodge 1-Laplacian $\tilde{\mathbf{L}_1}$, Schaub et al. [41] introduce a lifted space of edges, where the original $m = |S_1|$ directed edges are lifted to $2m$ directed edges. For example, if $(i, j) \in S_1$, then we add $(j, i)$ to the lifted space. Consequently, the edge flow $\mathbf{f} \in \mathcal{C}^1$ expands to a larger space $\mathcal{D}^1$ where there are two orientations for each edge, $|\mathcal{D}^1| = 2|\mathcal{C}^1|$. The matrix representation for this lifting procedure is $\mathbf{V} = [+\mathbf{I}_m \quad -\mathbf{I}_m]^T \in \mathbb{R}^{2m \times m}$. Then the probability transition matrix for this lifted random walk corresponding to $\tilde{L}_1$ is $\hat{\mathbf{P}} : -\frac{1}{2}\tilde{\mathbf{L}_1}\mathbf{V}^T = \mathbf{V}^T\hat{\mathbf{P}}$. In practice, we also perform a simpler row-wise normalization over $\mathbf{L}_1$ to obtain another form of probability transition matrix.

Using $\hat{\mathbf{P}}$, we can construct an edge-level random walk-based PE method to enrich edge data by encoding structure information, analogous to node-level RWSE. We will also discuss some variations and simplified versions of the aforementioned random walk on 1-simplices and theoretically analyze their expressivity.

**EdgeRWSE.** Similar to node-level random walk, a well-defined edge-level random walk contains some structure information and can be used to facilitate edge data, namely edge-level positional encoding. While node-level positional encodings have been widely studied, the edge-level positional encoding is a nearly blank field.

Inspired by (node-level) RWSE, EdgeRWSE is based on edge-level random walk. A full version of EdgeRWSE is based on the full edge-level random walk as we have stated above and in [41]. For undirected graphs, two edges with opposite directions $(i, j)$ and $(j, i)$ are again merged by summing

the two probabilities, that is, the lifted space $\mathcal{D}^1$ is mapped back to $\mathcal{C}^1$. Generally speaking, PE can be based on any injection functions $\psi$ in $\hat{\mathbf{P}}$ and its powers.

$$\text{EdgeRWSE}(\hat{\mathbf{P}})_i = \psi([\hat{\mathbf{P}}^k]), k = 1, 2, ...K \tag{3}$$

where $K$ is the maximum steps we consider. One possible example is to encode the return probability of each edge, which is written $\text{EdgeRWSE}_{\text{ret}}(\hat{\mathbf{P}})_i = \psi([\hat{\mathbf{P}}^k_{ii}]), k = 1, 2, ...K$. If $\psi$ is well defined, the theoretical expressivity of the full EdgeRWSE above is able to break the 2-FWL bottleneck of node-level RWSE. In practice, we can apply neural networks like MLP or Transformer to encode $\hat{\mathbf{P}}^k$ and concatenate them with the original edge features. Then any standard GNN is applicable for downstream tasks. If the GNN is at least as powerful as 1-FWL, then the GNN with EdgeRWSE is strictly more powerful than 1-FWL and can distinguish some non-isomorphic graph pairs in which 2-FWL fails.

In addition to the edge-level random walk in the lifted space of 1-simplicials in [41], we further define two simplified versions of the edge-level random walk only through lower adjacency. We neglect the 2-simplices or the triangles in our simplified version random walk, i.e. we only consider the 1-down neighbors that share a 0-simplices (node). In this way, $\hat{\mathbf{P}}$ becomes $\mathbf{P}_{down}$. This simplification will lead to a theoretically weaker expressivity than using full $\hat{\mathbf{P}}$, which will be bounded by 2-FWL. However, this simplification is appropriate and beneficial for real-world data that contain a small number of triangles. We illustrate these two variations temporarily on undirected connected graphs without multiple edges and self-loops for simplicity.

The two variations of edge-level random walk via down-neighbors differ in whether two lower adjacent nodes of the edge have the same status. Concretely, the first type of edge-level random walk based on $\mathbf{P}_{down}$, which we define as *directed* 1-*down random walk* follows a two-stage procedure at every step. The walk first selects one of the two lower-adjacent nodes with equal probability 0.5 each, then moves towards the neighboring edges connected with the selected node with equal probabilities. If there are no other edges connected to the selected node, the walk returns to the original edge. On the other hand, the second type, which we denote as *undirected* 1-*down random walk*, chooses the two nodes $u, v$ with probabilities proportional to their degrees minus one (since we want to exclude the case of returning to $e$ itself). Consequently, the walk transits to all 1-down neighbors of the source edge with equal probabilities.

In a similar way as the full EdgeRWSE, we propose two simplified versions of EdgeRWSE based on directed 1-down and undirected 1-down random walk, both can be implemented in a rather flexible way. As a special case, the return probabilities of each edge after $k = 1, \ldots, K$ steps are encoded, but notice again that it is not the only implementation choice.

We conclude by summarizing the expressivity of EdgeRWSE.

**Theorem 5.1.** *Full EdgeRWSE can distinguish some non-isomorphic graphs that are indistinguish by* 2-*FWL. EdgeRWSE based on directed and undirected* 1-*down random walk are not more powerful than* 2-*FWL.*

## 5.2 Sign and basis invariant edge-level positional encoding

**Theoretical analysis of Hodge 1-Laplacian spectrum.** Recall that the unnormalized Hodge 1-Laplacian is $\mathbf{L}_1 = \mathbf{B}_1^T \mathbf{B}_1 + \mathbf{B}_2 \mathbf{B}_2^T = \mathbf{L}_{1,down} + \mathbf{L}_{up}$. Here, we analyze the theoretical properties of Hodge 1-Laplacian including its spectrum, which provides solid insights into our following designs.

Note that previous simplicial networks [12, 47, 8, 7] are orientation equivariant and permutation equivariant; thus, they can only be applied to simplicial complexes where all edges are directed. This is frustrating if we want to boost general learning on graphs rather than simplicial complexes alone. However, the spectral analysis of Hodge 1-Laplacian is applicable to undirected graphs. An important property of Hodge Laplacians is that their eigenvalues are invariant to permutation and orientation (if the simplices are oriented), thus they could be directly applied to analyze undirected graphs. Hence in this section, we temporarily omit discussion on permutation and orientation invariance since they naturally hold. Instead, we care more about the sign and basis invariance in the field of spectral analysis [32].

We can show that the nonzero eigenvalues of $\mathbf{L}_{1,down}$ are the same as $\mathbf{L}_{0,up}$ and hence $\mathbf{L}_0$. This implies that if there are no 2-simplicials (triangles), Hodge 1-Laplacian has the same nonzero

eigenvalues as Hodge 0-Laplacian. However, the corresponding eigenvectors still provide different information about the nodes and edges, respectively.

**Theorem 5.2.** *The number of non-zero eigenvalues of Hodge 1-Laplacian $L_1$ is not less than the number of non-zero eigenvalues of Hodge 0-Laplacian $L_0$.*

One direct conclusion is that graph isomorphism based on Hodge 1-Laplacian isospectra is strictly more powerful than Hodge 0-Laplacian. Here we draw a conclusion on the theoretical expressivity of the $L_1$ isospectra:

**Theorem 5.3.** $L_1$ *isospectra is incomparable with* 1-*FWL and* 2-*FWL.*

Rattan and Seppelt [39] show that the $L_0$ isospectra is strictly bounded by 2-FWL. The $L_1$ isospectra, through the introduction of 2-simplices (triangles), can distinguish some non-isomorphic graph pairs that are indistinguishable by 2-FWL. See Appendix C for detailed examples.

The zero eigenvalues of $L_1$ have some more important properties. Its multiplicity is the 1-th Betti number $\beta_1$, which is exactly the number of cycles (except triangles) in the graph. We further consider the eigenvectors of $L_1$, each eigenvector $\mathbf{u}_i$ of the eigenvalues $\lambda_i$ has a length $m$, and each element $\mathbf{u}_{ij}$ in it reflects the weight of the corresponding edge $e_j$ at this frequency $\lambda_i$. The absolute values of elements corresponding to the edges in cycles are non-zero, while the edges not in cycles have zero weights in the eigenvectors. In other words, the eigenvectors of zero eigenvalues can efficiently mark the edges that are in a cycle. More intuitive illustration and theoretical proof are given in Appendix C.2.2.

**Hodge1Lap: sign and basis invariant edge PE.** In this section, we propose Hodge1Lap, a novel edge-level positional encoding method based on the spectral analysis of Hodge 1-Laplacian. To the best of our knowledge, this is the first sign and basis invariant edge-level PE based on Hodge 1-Laplacian $L_1$.

Recall the geometric meaning of the Hodge 1-Laplacian spectra in Section 5.2. Zero eigenvalues and eigenvectors reflect the cycles in the graph. These insights of Hodge 1-Laplacian spectra shed light on our design for edge-level positional encoding. Denote the eigenvalues $\lambda_i$ with multiplicity $m(i)$ as $\lambda_{i(1)}, \lambda_{i(2)}, \ldots, \lambda_{i(m_i)}$, respectively. The corresponding eigenvectors are $\mathbf{u}_{i(1)}, \ldots, \mathbf{u}_{i(m_i)}$, but note that these eigenvectors are: (i) not sign invariant, since if $L_1 \mathbf{u}_{i(j)} = 0, j = 1, ..., m_i$, then $L_1(-\mathbf{u}_{i(j)}) = 0$; (ii) not basis invariant if $m_i > 1$, since any $m_i$ linearly independent basis of the kernel space are also eigenvectors, and the subspace they span is identical to the kernel space. This is analogous to the $L_0$ eigenvectors: they are not sign and basis invariant, which makes it difficult for us to design sign and basis invariant positional encodings. Therefore, we propose a novel projection-based method to build Hodge1Lap, a sign and basis invariant edge-level positional encoding.

Formally, Hodge1Lap processes the eigenvalues $\lambda_i$ with multiplicity $m_i$ and relevant eigenvectors as follows. Recall the projection matrix

$$P_{proj,i} = \mathbf{U}\mathbf{U}^T = \sum_{j=1}^{m_i} \mathbf{u}_{i(j)} \mathbf{u}_{i(j)}^T \tag{4}$$

where the subscript $_{proj}$ is used to distinguish the projection matrix from probability transition matrix $P$, and $\mathbf{U} = [\mathbf{u}_{i(1)}, \ldots, \mathbf{u}_{i(m_i)}]$. For any vector $\mathbf{v} \in \mathbb{R}^m$, $P_{proj,i}\mathbf{v}$ projects it into the subspace spanned by the eigenvectors $u_{i(j)}, j = 1, \ldots, m_i$. It is straightforward to verify that the projection in the subspace is independent of the choice of basis $u_{i(j)}$ as long as they are linearly independent and hence is both sign and basis invariant. As long as the preimage $\mathbf{v}$ is well defined (e.g., permutation equivariant to edge index), the projection can satisfy permutation equivariance as well as sign and basis invariance. In Hodge1Lap, we propose to use two different forms of preimages: a unit vector $\mathbf{e} \in \mathbb{R}^m$ with each element $\mathbf{e}_j = \frac{1}{\sqrt{m}}$, and the original edge feature $\mathbf{X}(E) \in \mathbb{R}^{m \times d}$. The first variant considers puer structure information, while the second variant jointly encodes structure and feature information. Taking the first variant as an example, Hodge1Lap implemented by projection can be formulated as

$$\text{Hodge1Lap}_{\text{proj}}(E) = \sum_i \phi_i(P_{proj,i}\mathbf{e}) \tag{5}$$

where $\phi_i$ are injective functions and can be replaced by MLP layers, and the summation is performed over the interested eigen-subspaces.

In addition to the projection-based implementation of Hodge1Lap, we also implement other variants (analogously to the implementation of LapPE [30]): (i) We use a shared MLP $\phi$ to directly embed the $n_{eigen}$ eigenvectors corresponding to the smallest $n_{eigen}$ eigenvalues, where $n_{eigen}$ is a hyper-parameter shared for all graphs. We refer this implementation as $\mathrm{Hodge1Lap_{sim}}(E) = \sum_{i=1}^{n_{eigen}} \phi(\mathbf{u}_i)$. (ii) We take the absolute value of each element in eigenvectors before passing them to the MLP, which we denote as $\mathrm{Hodge1Lap_{abs}}(E) = \sum_{i=1}^{n_{eigen}} \phi(|\mathbf{u}_i|)$, where $|\cdot|$ means taking element-wise absolute value. It is remarkable that, while $\mathrm{Hodge1Lap_{proj}}$ is sign-invariant and basis-invariant, $\mathrm{Hodge1Lap_{sim}}$ is not invariant to both sign and basis, and $\mathrm{Hodge1Lap_{abs}}$ is sign-invariant yet not basis-invariant. We also allow combination of the above implementations; see Appendix E for more implementation details.

Our Hodge1Lap has elegant geometric meanings thanks to the spectral properties of $L_1$. For example, the kernel space of $L_1$ related to the zero eigenvalues is fully capable of **detecting cycles and rings** in graphs [23], which can play a significant role in many domains. In molecular graphs, for example, cycle structures such as benzene rings have crucial effects on molecular properties. Hodge1Lap is able to extract such rings in a natural way rather than manually listing them, and $\mathrm{Hodge1Lap_{abs}}$ is able to differentiate edges from distinct cycles. Intuitively, according to the Hodge decomposition theorem, any vector field defined on edges $\mathcal{C}^1$ can be decomposed into three orthogonal components: a solenoidal component, a gradient component and a harmonic (both divergence-free and curl-free) component; see Appendix A. $\ker(\mathbf{L}_1)$ is the harmonic component, and since divergence-free and curl-free edge flows can only appear on cycles, the eigenvectors corresponding to $\ker(\mathbf{L}_1)$ therefore mark out the cycles in the graph; see Appendix C.2.2 for more technical details and illustrative examples. Moreover, taking into account more subspaces other than the kernel space of $L_1$, Hodge1Lap contains other structure information since the eigenvectors are real and continuous vectors. Ideally, one can apply any sign and basis invariant functions to obtain a universal approximator [32] for functions on 1-faces besides projections, see Section 6 for general conclusions.

# 6 Random walk on higher-order and inter-order simplices

In Section 4 and Section 5, we systematically analyze the random walk and Hodge Laplacian-based PE and SE on 0-simplices (node level) and 1-simplices (edge level), respectively. As we have shown, introducing higher-order simplices into random walk benefits their theoretical expressivity. In this section, we formally introduce random walks on higher-order simplices and analyze their expressivity. We will also investigate the spectral analysis of Hodge $k$ Laplacians, whose normalization forms are closely related to random walks on $k$-simplices. Besides random walk within same-order simplices, we define a novel inter-order random walk that is able to transmit within different orders of simplices. This random walk scheme incorporates and unifies a wide range of simplicial networks [12, 8, 14].

## 6.1 Higher-order Hodge Laplacians and random walk

The $k$-th order Hodge Laplacian is defined as $\mathbf{L}_k = \mathbf{B}_k^* \mathbf{B}_k + \mathbf{B}_{k+1} \mathbf{B}_{k+1}^* = \mathbf{L}_{k,down} + \mathbf{L}_{k,up}$. Analogous to $\mathbf{L}_1$, a properly normalized Hodge $k$ Laplacian $\tilde{\mathbf{L}}_{\mathbf{k}}$ corresponds to a $k$-th order random walk on $k$-simplices in the lifted space. The matrix representation for the lifting is $\mathbf{V}_k = [+\mathbf{I}_{n_k} \quad -\mathbf{I}_{n_k}]^T \in \mathbb{R}^{2n_k \times n_k}$, where $n_k = |S_k|$ is the number of $k$-simplices in the simplicial complex $\mathcal{K}$. For undirected graphs, one only needs to sum over different orientations to get the cochain group $\mathcal{C}^k$ from $\mathcal{D}^k$, where $|\mathcal{D}^k| = 2|\mathcal{C}^k|$ is the cochain group in the lifted space. The transition matrix $\hat{\mathbf{P}}_k$ for $k$-th order random walk is defined through $-\frac{1}{2}\tilde{\mathbf{L}}_{\mathbf{k}}\mathbf{V}_k^T = \mathbf{V}_k^T \hat{\mathbf{P}}_k$.

Similarly to the edge-level random walk in the lifted space, the transition matrix $\hat{\mathbf{P}}_k$ describes that each step of $k$-th order random walk move towards either $k$-down neighbors or $k$-up neighbors. When going through the upper adjacent $k+1$ faces, the walk uniformly transits to an upper adjacent $k$-simplex with different orientation relative to the shared $k+1$ face, unless it has no upper adjacent faces. If the step is taken towards a lower-adjacent $k-1$ face, the walk transits along or against the original direction to one of its $k$-down neighbors.

Based on $\hat{\mathbf{P}}_k$, we can design $k$-th order RWSE for $k$-simplicial data according to the $k$-th order random walk, $k - \mathrm{RWSE} = \psi_k(\hat{\mathbf{P}}_k)$, where $\psi_k$ is an injective function that acts on either $\hat{\mathbf{P}}_k$ or its polynomials. If we maintain all $k$-RWSE for $k = 0, 1, \ldots, K$ in a simplicial complex $\mathcal{K}$ with

dimension larger than $K$, then we can get a more powerful algorithm by adding $K + 1$-RWSE to the $K + 1$-simplices in $\mathcal{K}$.

In addition to directly making use of the random walk on the $k$-simplices, spectral analysis of $\mathbf{L}_k$ also sheds light on PE designs for higher-order simplicial data. Based on the eigenvalues and eigenvectors of $\mathbf{L}_k$, we can build permutation equivariant and basis invariant functions defined on $\mathcal{K}_{k+1}$ that can simulate arbitrary $k$-cochain or $k$-form. Concretely, if we use the normalized version of $k$-th Hodge Laplacian $\Delta_k$ as in [24], the eigenvalues of $\Delta_k$ will be compact $0 \leq \lambda \leq k + 2$. Then applying a permutation equivariant and basis-invariant function such as *Unconstrained BasisNet* [32] on the eigenvalues and eigenvectors, we are able to approximate any $k$-form which is basis-invariant. We refer interested readers to Appendix C.3 for more details.

## 6.2   Inter-order random walk

The concept of random walk can be even generalized to a more universal version, which we denote as inter-order random walk. In each step, the inter-order random walk at a $k$-simplex can transit not only to the $k$-down neighbors and $k$-up neighbors (they are all $k$-simplices as well), but also to lower adjacent $k - 1$-simplices and upper adjacent $k + 1$-simplices. Here we denote the (unnormalized) adjacent matrix for the inter-order random walk on a $K$-order simplicial complex $\mathcal{K}$ as $\mathcal{A}_K(\mathcal{K})$, which is defined as

$$\mathcal{A}_K(\mathcal{K}) = \begin{bmatrix} \mathbf{L}_0 & \mathbf{B}_1 & & & & \\ \mathbf{B}_1^T & \mathbf{L}_1 & \mathbf{B}_2 & & & \\ & \cdots & \cdots & \cdots & & \\ & & \cdots & \cdots & \cdots & \\ & & & \mathbf{B}_{K-1}^T & \mathbf{L}_{K-1} & \mathbf{B}_K \\ & & & & \mathbf{B}_K^T & \mathbf{L}_K \end{bmatrix} \tag{6}$$

which is a block matrix with $\mathbf{L}_k$ in the $k$-th diagonal block, $\mathbf{B}_k^T$ and $\mathbf{B}_{k+1}$ in the offset $\pm 1$ diagonal blocks, while all other blocks are zeros. Although Chen et al. [14] also mentioned a similar block matrix, they do not pose a concrete form of the off-diagonal blocks. The inter-order adjacent matrix we define has a clear physical interpretation that one can only transform to simplices with different orders that are boundaries and co-boundaries of current simplex. A properly normalized version $\tilde{\mathcal{A}}_K$ can describe the inter-order random walk with a certain rule. Here, we give a property of the power of $\mathcal{A}_K$ which still holds in normalized versions.

$$\mathcal{A}_K^r = \begin{bmatrix} p_r(\mathbf{L}_0) & q_{r-1}(\mathbf{L}_{0,up})\mathbf{B}_1 & & & \\ q_{r-1}(\mathbf{L}_{1,down})\mathbf{B}_1^T & p_r(\mathbf{L}_1) & q_{r-1}(\mathbf{L}_{1,up})\mathbf{B}_2 & & \\ & \cdots & \cdots & \cdots & \\ & \cdots & \cdots & \cdots & \\ & & & q_{r-1}(\mathbf{L}_{K,down})\mathbf{B}_K^T & p_r(\mathbf{L}_K) \end{bmatrix} \tag{7}$$

where $p_r(\cdot)$ and $q_r(\cdot)$ are polynomials with maximum order $r$. The above equation states that simplices with differences of order larger than one cannot directly exchange information even after infinite rounds, but they can affect each other through the coefficients in $p_r$ and $q_{r-1}$ in the blocks on the offset $\pm 1$-diagonal blocks.

Several previous works such as [8] can be unified by $\mathcal{A}_K$. Additionally, we can make use of $\mathcal{A}_K^r$ to build random walk-based positional encoding for all simplices in the $K$-dimensional simplicial complex that contains rich information.

## 7   Experiments

In this section, we present a comprehensive ablation study on Zinc-12k to investigate the effectiveness of our proposed methods. We also verify the performance on graph-level OGB benchmarks. Due to the limited space, experiments on synthetic datasets and more real-world datasets as well as experimental details are presented in Appendix E.

**Ablation study on Zink-12k.**   Zinc-12k [17] is a popular real-world dataset containing 12k molecules. The task is the graph-level molecular property (constrained solubility) regression. In our ablation study, we use GINE [25], GAT [45], PNA [15], SSWL+ [50], GPS [38] and GRIT [33] as our base models, where the first three are message-passing based GNNs, SSWL+ is an instance of subgraph GNN, while GPS and GRIT are recent SOTA graph transformers. Four different factors are studied: (1) the node-level PE or SE, where RWSE refers to [18], LapPE refers to [30] and

Table 1: Ablation on Zinc-12k dataset [17] (MAE ↓). Highlighted are the first, second results.

| model | Node PE/SE | EdgeRWSE | Hodge1Lap | RWMP | Test MAE |
|---|---|---|---|---|---|
| GIN [46] | - | - | - | - | $0.526 \pm 0.051$ |
| GSN [9] | - | - | - | - | $0.101 \pm 0.010$ |
| Graphormer [48] | - | - | - | - | $0.122 \pm 0.006$ |
| SAN [30] | - | - | - | - | $0.139 \pm 0.006$ |
| GIN-AK+ [53] | - | - | - | - | $0.080 \pm 0.001$ |
| CIN [7] | - | - | - | - | $0.079 \pm 0.006$ |
| Specformer [6] | - | - | - | - | $0.066 \pm 0.003$ |
| GINE [25] | - | - | - | - | $0.133 \pm 0.002$ |
| GINE | - | directed | - | - | $0.110 \pm 0.003$ |
| GINE | - | undirected | - | - | $0.104 \pm 0.008$ |
| GINE | - | - | abs | - | $0.102 \pm 0.004$ |
| GINE | - | - | project | - | $0.091 \pm 0.004$ |
| GINE | LapPE | - | - | - | $0.120 \pm 0.005$ |
| GINE | RWSE | - | - | - | $0.074 \pm 0.003$ |
| GINE | RWSE | directed | - | - | $0.070 \pm 0.003$ |
| GINE | RWSE | undirected | - | - | $0.069 \pm 0.002$ |
| GINE | RWSE | - | abs | - | $0.068 \pm 0.003$ |
| GINE | RWSE | - | project | - | $0.068 \pm 0.004$ |
| GINE | RWSE | - | - | True | $0.068 \pm 0.003$ |
| GINE | RWSE | - | project | True | $0.066 \pm 0.003$ |
| GINE | RWSE | Full-EdgeRWSE | - | - | $0.069 \pm 0.003$ |
| GINE | Inter-RWSE | Inter-RWSE | - | - | $0.083 \pm 0.006$ |
| GINE | RWSE | Cellular | - | - | $0.068 \pm 0.003$ |
| GAT [45] | - | - | - | - | $0.384 \pm 0.007$ |
| GAT | - | undirected | - | - | $0.163 \pm 0.008$ |
| GAT | - | - | project | - | $0.130 \pm 0.005$ |
| PNA [15] | - | - | - | - | $0.188 \pm 0.004$ |
| PNA | - | undirected | - | - | $0.104 \pm 0.004$ |
| PNA | - | - | project | - | $0.074 \pm 0.005$ |
| SSWL+ [50] | - | - | - | - | $0.070 \pm 0.005$ |
| SSWL+ | - | undirected | - | - | $0.067 \pm 0.005$ |
| SSWL+ | - | - | project | - | $0.066 \pm 0.003$ |
| GPS [38] | - | - | - | - | $0.113 \pm 0.005$ |
| GPS | RWSE | - | - | - | $0.070 \pm 0.004$ |
| GPS | RWSE | undirected | - | - | $0.068 \pm 0.004$ |
| GPS | RWSE | - | project | - | $0.064 \pm 0.003$ |
| GRIT [33] | - | - | - | - | $0.149 \pm 0.008$ |
| GRIT | RWSE | - | - | - | $0.081 \pm 0.010$ |
| GRIT | SPDPE | - | - | - | $0.067 \pm 0.002$ |
| GRIT | RDPE | - | - | - | $0.059 \pm 0.003$ |
| GRIT | RRWP | - | - | - | $0.059 \pm 0.002$ |
| GRIT | - | undirected | - | - | $0.103 \pm 0.006$ |
| GRIT | - | - | project | - | $0.086 \pm 0.005$ |
| GRIT | RRWP | undirected | - | - | $0.058 \pm 0.002$ |
| GRIT | RRWP | - | project | - | $0.057 \pm 0.003$ |

"-" suggests no node-level PE/SE; (2) EdgeRWSE, the edge-level SE based on spatial domain of 1-down random walk, where "directed" and "undirected" are used to distinguish the two types of simplified version of 1-down random walk; (3) Hodge1Lap, the edge-level PE based on spectra of $\mathbf{L}_1$, where "abs" refers to the sign-invariant method (summing over absolute values of eigenvectors, or $\mathrm{Hodge1Lap_{abs}}$), and "project" refers to the sign and basis invariant method (project the unit vector into interested subspace, or $\mathrm{Hodge1Lap_{proj}}$); (4) RWMP, a novel Random Walk Message Passing scheme we propose, which performs message passing based on probability calculated by a distance metric; see Appendix D for details of RWMP.

Table 2: Experiments on graph-level OGB benchmarks [26]. Highlighted are the first, second, **third** test results.

| model | ogbg-molhiv (AUROC $\uparrow$) | ogbg-molpcba (Avg. Precision $\uparrow$) |
|---|---|---|
| GIN+virtual node | $0.7707 \pm 0.0149$ | $0.2703 \pm 0.0023$ |
| GSN (directional) | $0.8039 \pm 0.0090$ | - |
| PNA | $0.7905 \pm 0.0132$ | $0.2838 \pm 0.0035$ |
| SAN | $0.7785 \pm 0.2470$ | $0.2765 \pm 0.0042$ |
| GIN-AK+ | $0.7961 \pm 0.0110$ | $0.2930 \pm 0.0044$ |
| CIN | $0.8094 \pm 0.0057$ | - |
| GPS | $0.7880 \pm 0.0101$ | $0.2907 \pm 0.0028$ |
| Specformer | $0.7889 \pm 0.0124$ | $0.2972 \pm 0.0023$ |
| GPS+EdgeRWSE | $0.7891 \pm 0.0118$ | **$0.2934 \pm 0.0025$** |
| GPS+Hodge1Lap | **$0.8021 \pm 0.0154$** | $0.2937 \pm 0.0023$ |

The full results of performance on the Zinc dataset are reported in Table 1. Note that all our base models are improved when augmented with our EdgeRWSE or Hodge1Lap: both GAT and PNA reduce by over $50\%$ MAE. In particular, a simple GINE without using any transformer or subgraph GNN variations is able to surpass GPS with our PE/SE, verifying the impressive effectiveness of our proposed methods. Applying EdgeRWSE and Hodge1Lap to GRIT results in new **State-of-the-Art** performance. Regarding ablation, all variants of our EdgeRWSE and Hodge1Lap can improve performance of base models, see Appendix E for more implementation details of these variants. One may observe that RWSE is significantly beneficial in this task, and combining node-level RWSE and our edge-level PE/SE methods would lead to a further performance gain. In general, Hodge1Lap shows better performance than EdgeRWSE, indicating the effectiveness of embedding structures such as rings through spectral analysis. The effect of whether EdgeRWSE is directed or the implementation method in Hodge1Lap is rather small. We also observe that Full-EdgeRWSE, Inter-RWSE, and CellularRWSE are beneficial, see Appendix E for more details. Additionally, the RWMP mechanism is also capable of improving performance, which we will analyze in Appendix D.

**Experiments on OGB benchmarks.** We also verify the performance of EdgeRWSE and Hodge1Lap on graph-level OBG benchmarks, including the ogbg-molhiv and ogbg-molpcba datasets. The results are shown in Table 2. We apply our Hodge1Lap and EdgeRWSE to both GatedGCN and GPS(consists of GatedGCN and Transformer) and show that our methods can improve both architectures. In general, both two edge-level PE/SE are able to achieve comparable performance as the SOTA models, though EdgeRWSE suffers from overfitting on ogbg-molhiv. It should be noted that SOTA results on ogbg-molhiv typically involve manually crafted structures, including GSN [9] and CIN [7]. Natural methods and complex models usually suffer from overfitting and cannot generalize well in the test set.

## 8  Conclusions

In this paper, we propose to facilitate graph neural networks through the lens of random walk on simplicial complexes. The random walk on $k$-th order simplices is closely related to Hodge $k$ Laplacian $\mathbf{L}_k$, and we emphasize that both spatial analysis of random walk and spectra of $\mathbf{L}_k$ can improve the theoretical expressive power and performance of GNNs. For 0-simplices, we connect a number of exsiting PE and SE methods (such as RWSE) via node-level random walk, and further provide a theoretical expressivity bound. For 1-simplices, we propose two novel edge-level PE and SE methods, namely EdgeRWSE and Hodge1Lap. EdgeRWSE directly encodes information based on edge-level random walk, while Hodge1Lap is the first sign and basis invariant edge-level PE based on Hodge-1 Laplacian spectra. We also generalize our theory to arbitrary-order simplices, showing how $k$-order and inter-order random walk as well as spectral analysis of Hodge Laplacians can facilitate graph and simplicial learning. Besides analyzing theoretical expressive power and physical meanings of these random walk-based methods, we also verify the effectiveness of our methods, which achieve SOTA or highly competitive performance on several datasets.

## Acknowledgments and Disclosure of Funding

The work is supported in part by the National Natural Science Foundation of China (62276003), the National Key Research and Development Program of China (No. 2021ZD0114702), and Alibaba Innovative Research Program.

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

# A Discrete Hodge-deRham theory of abstract simplicial complex

Inspired by differential geometry and algebraic topology, this work investigates how random walk on simplicial complexes can facilitate graph and simplicial learning. In our main text, we merely introduce the Hodge Laplacian (in Hilbert space) due to limited space. In this section, we give a complete background on discrete Hodge-deRham theory of abstract simplicial complexes to help readers better understand relevant concepts.

An abstract simplicial complex $\mathcal{K}$ in a finite set $V$ is a collection of subsets of $V$ that is closed under inclusion. In our paper, $V$ will be a vertex set $[n] = \{1, 2, ..., n\}$ if without special statement. An element of cardinality $k + 1$ is called a $k$-face or $k$-simplex of $\mathcal{K}$. For instance, 0-faces are usually called vertices, 1-faces are directed edges and 2-faces are 3-cliques (triangles) with an orientation. We denote the collection of all $k$-faces of $\mathcal{K}$ as $S_k(\mathcal{K})$. The dimension of a $k$-face is $k$ and the dimension of a complex $\mathcal{K}$ is defined as the maximum dimension of faces in $\mathcal{K}$.

The definition of neighbors of simplices is crucial in this paper. Two $k + 1$-simplices sharing a collective $k$-face are called $k$-down neighbors, and two $k$-simplices sharing a collective $k + 1$-simplex are called $k + 1$-up neighbors. Generally, a face $F$ is chosen an ordering on its vertices and is said to be oriented, denoted by $[F]$. For any permutation element $\sigma \in \mathcal{G}_{k+1}$ where $\mathcal{G}_{k+1}$ is the symmetric group of permutations on $\{0, ..., k\}$, two orderings of vertices transformed by $\sigma$ are said to determine the same orientation if $\sigma$ is an even permutation and opposite if $\sigma$ is odd. In addition, a $k$-cochain or $k$-form is a function defined on $\mathcal{K}_{k+1}$, $f : V \times \cdots \times V \to \mathbb{R}$ that satisfies the following.

$$f(i_{\sigma(0)}, \ldots, i_{\sigma(k)}) = \text{sgn}(\sigma) f(i_0, \ldots, i_k) \tag{8}$$

for all $\{i_0, \ldots, i_k\} \in \mathcal{K}_{k+1}$ and all $\sigma \in \mathcal{G}_{k+1}$. Specifically, $f(i_0, \ldots, i_k) = 0$ if $\{i_0, \ldots, i_k\} \notin \mathcal{K}_{k+1}$. Although they have the structure of vector spaces, the vector spaces are usually called cochain groups $\mathcal{C}^k(\mathcal{K}, \mathbb{R})$. Chain groups $\mathcal{C}_k(\mathcal{K}, \mathbb{R})$ are defined as duals of co-chain groups. In addition, we define the simplicial coboundary maps $\delta_k : \mathcal{C}^k(\mathcal{K}, \mathbb{R}) \to \mathcal{C}^{k+1}(\mathcal{K}, \mathbb{R})$:

$$(\delta_k f)([v_0, \ldots, v_{i+1}]) = \sum_{j=0}^{k+1} (-1)^j f([v_0, \ldots, \hat{v}_j, \ldots, v_{k+1}]) \tag{9}$$

where $\hat{v}_j$ suggests that the vertex $v_j$ is omitted. One can view $\delta_k$ as the dual of the boundary map $\partial_{k+1}$, which connects the cochain complex of $\mathcal{K}$ with coefficients in $\mathbb{R}$. Further, we can define the adjoint of coboundary operator: $\delta_k^* : \mathcal{C}^{k+1}(\mathcal{K}, \mathbb{R}) \to \mathcal{C}^k(\mathcal{K}, \mathbb{R})$. Therefore, we have the following connection.

$$\mathcal{C}^{k+1}(\mathcal{K}, \mathbb{R}) \underset{\delta_k^*}{\overset{\delta_k}{\leftrightarrows}} \mathcal{C}^k(\mathcal{K}, \mathbb{R}) \underset{\delta_{k-1}^*}{\overset{\delta_{k-1}}{\leftrightarrows}} \mathcal{C}^{k-1}(\mathcal{K}, \mathbb{R}) \tag{10}$$

After determining the inner products $<, >$, we can define the adjoint of the coboundary operator $\delta_k^* : \mathcal{C}^{k+1}(\mathcal{K}, \mathbb{R}) \to \mathcal{C}^k(\mathcal{K}, \mathbb{R})$

$$< \delta_k \mathbf{f}_1, \mathbf{f}_2 >_{\mathcal{C}^{k+1}} = < \mathbf{f}_1, \delta_k^* \mathbf{f}_2 >_{\mathcal{C}^k} \tag{11}$$

where $\mathbf{f}_1 \in \mathcal{C}^k(\mathcal{K}, \mathbb{R})$, $\mathbf{f}_2 \in \mathcal{C}^{k+1}(\mathcal{K}, \mathbb{R})$ are arbitrary. Specifically, in the Hilbert space $L^2$, the matrix representations of $\delta_k$ and $\delta_k^*$ are adjoint matrix $\mathbf{B}_{k+1}^*$ and $\mathbf{B}_{k+1}$ as described in the main texts.

Define the Hodge $k$-Laplacian operator (also called combinatorial Laplace operator):

$$\mathbf{L}_k = \mathbf{L}_{k,down} + \mathbf{L}_{k,up} = \delta_{k-1} \delta_{k-1}^* + \delta_k^* \delta_k \tag{12}$$

where we omit the notation $\mathcal{K}$ for simplicity. It is easy to verify by definition that all three operators $\mathbf{L}_k, \mathbf{L}_{k,up}, \mathbf{L}_{k,down}$ are self-adjoint, nonnegative and compact.

In the Hilbert space, the matrix representations for boundary and co-boundary operators are adjacent matrices of the $k$ and $k + 1$ order simplices. In order to keep coordinate with most existing literature,

we write the matrix representation for $\delta_k^*$ as $\mathbf{B}_{k+1} \in \mathbb{R}^{|S_k| \times |S_{k+1}|}$ (one can view it as the adjacent matrix of $k$-th and $k+1$-th simplices); therefore, we have the definition in our main text:

$$\mathbf{L}_k = \mathbf{B}_k^* \mathbf{B}_k + \mathbf{B}_{k+1} \mathbf{B}_{k+1}^* \tag{13}$$

where $\mathbf{B}_k^* = \mathbf{B}_k^T$ is the adjoint of $\mathbf{B}_k$ and is equivalent to the transpose of $\mathbf{B}_k$ in the Hilbert space. It is remarkable that when $k = 0$, $\mathbf{L}_0$ is exactly the graph Laplacian $\mathbf{L}_0 = \mathbf{D} - \mathbf{A}$. In our paper, we make use of higher-order Hodge Laplacians such as $\mathbf{L}_1$ rather than previously used $\mathbf{L}_0$ alone.

Note that the following equation always holds:

$$\delta_k \delta_{k-1} = 0 \tag{14}$$

This result is sometimes called the *fundamental theorem of topology*, which can be intuitively interpreted as "the coboundary of a coboundary is zero".

The kernel space of $\mathbf{L}_k$ is called the $k$-th cohomology group:

$$\tilde{\mathcal{H}}^k(\mathcal{K}, \mathbb{R}) := \ker(\delta_k)/\mathrm{im}(\delta_{k-1}) \cong \ker(\delta_k) \cap \ker(\delta_{k-1}^*) = \ker(\mathbf{L}_k) \tag{15}$$

We will write $\tilde{\mathcal{H}}^k(\mathcal{K}, \mathbb{R})$ simply as $\tilde{\mathcal{H}}^k$ without causing confusion. The kernel space is closely associated with harmonic functions and will play an important role in our following analysis.

*Proof.* We already have $\delta_k \delta_{k-1} = 0$ and thus $\delta_{k-1}^* \delta_k^* = 0$. Then,

$$\mathrm{im}(\mathbf{L}_{k,\mathrm{down}}) \subset \ker(\mathbf{L}_{k,\mathrm{up}}) \mathrm{im}(\mathbf{L}_{k,\mathrm{up}}) \subset \ker(\mathbf{L}_{k,\mathrm{down}}) \tag{16}$$

Therefore,

$$\ker(\mathbf{L}_k) = \ker(\delta_k^* \delta_k) \cap \ker(\delta_{k-1}\delta_{k-1}^*) \tag{17}$$
$$= \ker(\delta_k) \cap \ker(\delta_{k-1}^*) \tag{18}$$
$$= \ker(\delta_k) \cap \left(\mathrm{im}(\delta_{k-1})\right)^{\perp} \tag{19}$$
$$\cong \tilde{\mathcal{H}}^k \tag{20}$$

$\square$

Equation (16) reveals an important conclusion: $\lambda > 0$ is a nonzero eigenvalue of $\mathbf{L}_k$ if and only if it is a nonzero eigenvalue of $\mathbf{L}_{k,up}$ and $\mathbf{L}_{k,down}$. Let the sorted eigenvalues of operator $\mathbf{A}$ be $\mathbf{s}(\mathbf{A}) = (\lambda_0, \ldots, \lambda_m)$, where the eigenvalues are in a weakly increasing rearrangement. Denote $\mathbf{s}(\mathbf{A}) \doteq \mathbf{s}(\mathbf{B})$ if the multisets of $\mathbf{s}(\mathbf{A})$ and $\mathbf{s}(\mathbf{B})$ have exactly the same nonzero eigenvalues (or they only differ in the multiplicities of zero). Then we have

$$\mathbf{s}(\mathbf{L}_k) \doteq \mathbf{s}(\mathbf{L}_{k,up}) \cup \mathbf{s}(\mathbf{L}_{k,down}) \tag{21}$$

Further using $\mathbf{s}(AB) \doteq \mathbf{s}(BA)$, we have an important property that bridges the up and down Hodge Laplacians of the adjacent dimension:

$$\mathbf{s}(\mathbf{L}_{k,up}) \doteq \mathbf{s}(\mathbf{L}_{k+1,down}) \tag{22}$$

Now we turn our attention to the zero eigenvalues and eigenvectors. The multiplicity of zero eigenvalues of $\mathbf{L}_k$, or the dimension of null space of Hodge $k$-Laplacian $\ker(\mathbf{L}_k)$, is called the $k$-th Betti number $\beta_k$. This is exactly the number of cycles composed of $k$-simplicials that are not induced by a $k$-boundary, or intuitively, $k$-dimensional "holes" in the simplicial complex $\mathcal{K}$. For example, zero eigenvalues and their eigenvectors of $\mathbf{L}_0$ are associated with 0-th homology group of the graph,

corresponding to the connected components in the graph. The zero eigenvalues and eigenvectors of $\mathbf{L}_1$ are associated with cycles (in the usual sense), and those of $\mathbf{L}_2$ correspond to cavities.

Also note that the following relation always holds.

$$\operatorname{im}(\mathbf{L}_k) = \operatorname{im}(\delta_k^*) \oplus \operatorname{im}(\delta_{k-1}) \tag{23}$$

Finally, we present the central theorem of Hodge theory, known as Hodge decomposition. It states that $\mathcal{C}^k$ can be decomposed into three orthogonal subspaces:

$$\mathcal{C}^k(\mathcal{K}, \mathbb{R}) = \operatorname{im}(\delta_k^*) \oplus \overbrace{\underbrace{\ker(\mathbf{L}_k)}_{\ker(\delta_k)} \oplus \operatorname{im}(\delta_{k-1})}^{\ker(\delta_{k-1}^*)} \tag{24}$$

For example, for a vector field $\mathcal{C}^1$, the above equation can be interpreted as follows. Any edge flow can be decomposed into three orthogonal components: a solenoidal (divergence-free) component, a harmonic (both divergence-free and curl-free) component, and a gradient (curl-free) component. In a discrete graph, a divergence of a vertex (0-simplex) is defined as the excess of outgoing over incoming weights, while the curl in a directed triangle (2-simplex) is defined as the edge sum around the triangle along the positive direction.

The Hodge decomposition theorem provides insight into our methods. For instance, the kernel space of $\mathbf{L}_1$ (which plays an important role in our Hodge1Lap) is both divergence-free and curl-free, thus only occurring along the cycles in the graph. Moreover, the Hodge decomposition holds for any order of cochains $\mathcal{C}^k$, enriching the physical insights of our PE and SE based on Hodge $k$-Laplacians.

## B  Classical results of node-level random walk

Here are some classical conclusions about the random walk at node level.

Let $X_k$ be the node we are at step $k$, and $S_k(x)$ be the number of times that the random walk visits $x$ during first $t$ steps, then we have

**Theorem B.1.** $\lim_{k \to \infty} \mathbb{E}[\frac{S_k(x)}{k}] = \frac{d(x)}{2m}$. $\frac{S_k(x)}{k}$ *tends to stationary probability $\pi$ in probability as* $k \to \infty$.

Denote the mean hitting time of $y$ from $x$ by $H_{(}x, y)$, then clearly

$$H(x, y) = \sum_{k=1}^{\infty} k\mathbb{P}(X_k = y, X_k \neq y, 1 \leq i < k \Big| X_0 = x) \tag{25}$$

Generally, $H(x, y)$ is not symmetric. Specifically, $H(x, x)$ is the mean return time from $x$ to itself. We have

$$H(x, x) = 1 + \sum_{z \in V} P_{xz} H(z, x) = 1 + \frac{1}{d(x)} \sum_{z \in \mathcal{N}(x)} H(z, x) \tag{26}$$

In terms of connected graph, we have the following conclusion.

**Theorem B.2.** *If the graph $G$ is connected, then $H(x, x) = \frac{2m}{d(x)}$*

A slight variation of the conclusion states,

**Theorem B.3.** $\frac{\mathbb{E}S_k(yx)}{k} = \frac{\mathbb{E}S_k(y)}{kd(y)} \to \frac{1}{2m}$

**Theorem B.4.** *$G$ is a connected graph, then a simple random walk satisfies* $\frac{1}{2m} \sum_{x \in V(G)} \sum_{y \in \mathcal{N}(x)} H(x, y) = n - 1$.

We have a reformulation for the above result if we define the mean commute time as $C(x, y) = H(x, y) + H(y, x)$, then

$$\frac{1}{2m} \sum_{(x,y) \in E(G)} C(x, y) = n - 1 \qquad (27)$$

There is an interesting and well-known connection between mean commute time and resistance between nodes $r_{xy}$, see [2, 51].

**Theorem B.5.** $C(x, y) = 2mr_{xy}$

In addition, the cover time is defined as $C := \max\{C_v : v \in V\}$, where $C_v$ is the expected number of steps taken by a random walk that starts from $v$ to hit every vertex in the graph. A bound is obtained by Aleliunas et al. (1979),

**Corollary B.6.** *The cover time $C$ is at most $2m(n - 1)$*

These classical conclusions in node-level random walk provide insights into the design of PE/SE methods such as resistance distance (RD). However, since they have been widely studied, we will not restate them.

# C   Theoretical analysis and proof

In this section, we provide proofs, details and examples for Section 4, Section 5 and Section 6 in the main text. Note that all the following results are novel and are of great significance in theoretically understanding our methods.

## C.1   Random walk on 0-simplices

### C.1.1   Theoretical analysis of 0-RWSE

We start with proving the theoretical expressive power of PE/SE based on spatical domain of random walk on 0-simplices. In order to distinguish RWSE for different orders of simplices, we denote the normal node-level RWSE as 0-RWSE, our proposed EdgeRWSE as 1-RWSE, and RWSE based on random walk on $k$-simplices as $k$-RWSE.

We first introduce 2-FWL, a powerful graph isomorphism test. It assigns colors to all 2-tuples of nodes and iteratively updates them. The initial color $c_2^0(\boldsymbol{v}, G)$ of tuple $\boldsymbol{v} \in V(G)^2$ is determined by the isomorphism type of tuple $\boldsymbol{v}$ [34]. At the $t$-th iteration, the color updating scheme is

$$c_2^t(\boldsymbol{v}, G) = \text{Hash}\Big(c_2^{t-1}(\boldsymbol{v}, G), \{\{$$

$$\big(c_2^{t-1}(\psi_i(\boldsymbol{v}, u), G) | i \in [k]\big) | u \in V(G)\}\}\Big), \quad (28)$$

where $\psi_i(\boldsymbol{v}, u)$ means replacing the $i$-th element in $\boldsymbol{v}$ with $u$. The color of $\boldsymbol{v}$ is updated by its original color and the color of its high-order neighbors $\psi_i(\boldsymbol{v}, u)$. The color of the whole graph is the multiset of all tuple colors,

$$c_2^t(G) = \text{Hash}(\{\{c_2^t(\boldsymbol{v}, G) | \boldsymbol{v} \in V(G)^2\}\}). \qquad (29)$$

Given two function $f, g$, $f$ can be expressed by $g$ means that there exists a function $\phi$ $\phi \circ g = f$, which is equivalent to given arbitrary input $H, G$, $f(H) = f(G) \Rightarrow g(H) = g(G)$. We use $f \rightarrow g$ to denote that $f$ can be expressed with $g$. If both $f \rightarrow g$ and $g \rightarrow f$, there exists a bijective mapping between the output of $f$ to the output of $g$, denoted as $f \leftrightarrow g$.

**Theorem C.1.** (Theorem 4.1. in main text.) 0-*RWSE is strictly less powerful than* 2-*FWL.*

*Proof.* First, we will prove that 2-FWL can express 0-RWSE. Specifically, we will prove that $\forall u \in V, c_2^{(t)}(uv, G) \rightarrow [(D^{-1}A)_{uv}^k | k = 1, 2, ..., t] \forall t = 1, 2, ...,$. We prove this by induction on $t$.

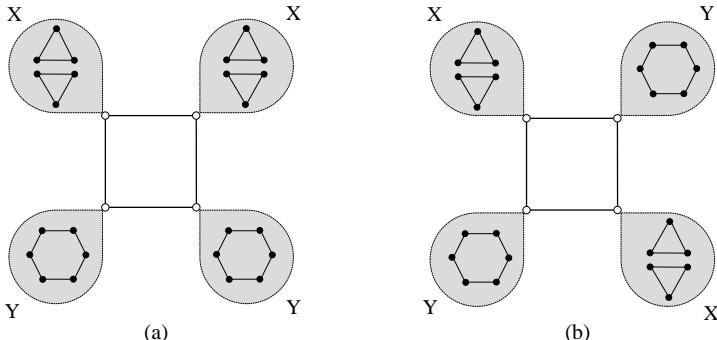

Figure 1: A pair of non-isomorphic graphs that are distinguishable by 2-FWL but indistinguishable by 0-RWSE in Theorem C.1. The shaded sectors denote bicliques $K_{1,6}$, i.e. all six solid nodes in the shade are connected with the hollow node in the square trunk.

1. First, 2-FWL can capture degree information in one iteration.

$$c_2^{(1)}(uv, G) \rightarrow c_2^{(0)}(uv, G), \{\{(c^{(0)}(uw, G), c^{(0)}(wv, G))|w \in V\}\} \tag{30}$$

$$\rightarrow c_2^{(0)}(uv, G), \{\{c^{(0)}(uw, G)|w \in G\}\} \qquad\qquad \rightarrow A_{uv}, D_u \tag{31}$$

$$\rightarrow (D^{-1}A)_{uv} \tag{32}$$

Therefore, $c_2^{(1)}(uv, G) \rightarrow (D^{-1}A)_{uv}$.

2. If $c_2^{(t)}(uv, G) \rightarrow [(D^{-1}A)_{uv}^k|k = 1, 2, ..., t]$, at $t + 1$-th 2-FWL iteration.

$$c_2^{(t+1)}(uv, G) \rightarrow c_2^{(t)}(uv, G), \{\{(c^{(t)}(uw, G), c^{(t)}(wv, G))|w \in V\}\} \tag{33}$$

$$\rightarrow [(D^{-1}A)_{uv}^k|k = 1, 2, ..., t], \{\{(D^{-1}A)_{uw}^k(D^{-1}A)_{wv}^k|w \in V\}\} \tag{34}$$

$$\rightarrow [(D^{-1}A)_{uv}^k|k = 1, 2, ..., t], \sum_w (D^{-1}A)_{uw}^k(D^{-1}A)_{wv}^k \tag{35}$$

$$\rightarrow [(D^{-1}A)_{uv}^k|k = 1, 2, ..., t + 1] \tag{36}$$

Next we prove the strictness, i.e. there exists some non-isomorphic graph pairs distinguishable by 2-FWL but not by RWSE. We will show that the graph pair in Figure 1 satisfies this requirement. Both graphs consist of a square backbone with 4 nodes (hollow in the figure), and each backbone node connects with every node in either a copy of $X$ or $Y$. The graph $X$ consists of two triangles, while the graph $Y$ consists of a six-ring. The difference of two graphs is that two backbone nodes connected with the same graph $X$ or $Y$ are adjacent in (a), but are not adjacent in (b). These two graphs are non-isomorphic and can be distinguished by 2-FWL, but not by RWSE.

The conclusion of 2-FWL can distinguish graphs (a) and (b) in Figure 1 can be proved by the 3-pebble bijective game [13] on these two graphs, see [39] for details.

Then we show that 0-RWSE fails to distinguish these two graphs step by step.

1. For the eight hollow nodes in the square backbones in (a) and (b), regardless of whether it is connected to a copy of $X$ or $Y$, once we are at the other six nodes in the biclique $K_{1,6}$ connected with it, the probability of returning to the backbone node for the first time within $k$ step is $P_k = \frac{1}{3}^k$. This can be easily verified since every solid node in $X$ and $Y$ has degree 3, therefore the steps it takes to return to the backbone is a geometric distribution parameterized by $\frac{1}{3}$.

2. Consider random walks only within the 6 solid nodes in biclique $K_{1,6}$ and not returning to the backbone. The probability distributions (or PMFs) for all nodes in all copies of $X$ are the same (which we denote as $p_{XX}$), and PMFs for all nodes in all copies of $Y$ are the same (which we denote as $p_{YY}$). The PMFs for nodes in $X$ and $Y$ are different ($p_{XX} \neq p_{YY}$). This is also straightforward to verify, since nodes in $X$ can only get to the other two solid nodes if they do not return to the

backbone, while nodes in $Y$ can get to all nodes. Therefore, RWSE can distinguish subgraphs $X$ and $Y$ (not including the connection between these 6 nodes with backbone).

3. For any backbone node in two graphs, it will have probability $\frac{3}{4}$ to walk toward the connected $X$ or $Y$ (but the walk cannot distinguish whether it is connected with $X$ or $Y$ according to 1.), and probability $\frac{1}{4}$ to walk towards another backbone connected with $X$ or $Y$ (again, either case is the same since the returning PMF to backbone cannot distinguish $X$ and $Y$). Combining 1. and 3. and using induction, we can conclude that the returns through biclique or backbone are the same in two graphs, and thus the return probabilities of all eight backbone nodes are the same, which we denote as $p_B$.

4. For all nodes in any copies of $X$, in each step $t$, the return PMF to $x \in X$ will have probability $\frac{2}{3}$ depending on $p_{XX}$, and probability $\frac{1}{3}$ depending on $p_B$. Once the connected backbone node is reached, the walk returns to the backbone node according to $p_B$, then returns to $X$ with probability $\frac{3}{4}$. Therefore, the final return probabilities of $x \in X$ are the same, depending on $p_{XX}$ and $p_B$. Similarly, the return probabilities of $y \in Y$ are the same, depending on $p_{YY}$ and $p_B$.

5. Finally, for both two graphs, all backbone nodes are assigned the same 0-RWSE feature as $\mathbf{b}$, all $x \in X$ are assigned the same $\mathbf{x}$, and all $y \in Y$ are assigned $\mathbf{y}$. Therefore, the two graphs have 4 $\mathbf{b}$, 12 $\mathbf{x}$ and 12 $\mathbf{y}$, indicating that 0-RWSE fails to distinguish these two graphs. $\qquad\square$

**Corollary C.2.** 1-*FWL with* 0-*RWSE initialization is strictly more powerful than* 0-*RWSE, but is not more powerful than* 2-*FWL.*

*Proof.* Figure 1 provides a pair of graphs that 1-FWL with 0-RWSE can differentiate, while 0-RWSE cannot. Therefore, 1-FWL with 0-RWSE initialization is strictly more powerful than 0-RWSE. However, as shown in Theorem C.1, $\exists t > 0, c^{(t)}(uu, G) \to RWSE_u$. Therefore,

$$c^{(t+1)}(uv, G) \to c^{(0)}(uv, G), \{(c^{(t)}(uw, G), c^{(t)}(wv, G))|w \in V\} \tag{37}$$

$$\to c^{(0)}(uv, G), \{c^{(t)}(uw, G)|w \in V\}, \{c^{(t)}(wv, G)|w \in V\} \tag{38}$$

$$\to c^{(0)}(uv, G), c^{(t)}(uu, G), c^{(t)}(vv, G) \tag{39}$$

$$\to c^{(0)}(uv, G), RWSE_u, RWSE_v \tag{40}$$

$$\to c^{(0)}(uv, G^{RWSE}). \tag{41}$$

where $G^{RWSE}$ is the graph with the 0-RWSE feature. In other words, 2-FWL can first capture RWSE with a few iterations and then simulate 1-FWL on the graph with RWSE. $\qquad\square$

Despite its upper bound of 2-FWL, 0-RWSE is capable of distinguishing some cases where 1-FWL fails. For example, the non-isomorphic graph pair shown in Figure 2 is a well-known case that 1-FWL does not distinguish. However, 0-RWSE can easily distinguish them, since a 0-random walk on graph (b) can only visit three nodes within the same triangle, thus having a larger return probability in the third step (and some subsequent steps) than on graph (a). 0-RWSE can also greatly enhance the performance of GNNs in real-world tasks, as shown in experiments.

For the same reason, we can prove that the newly proposed RRWP [33] SE is also strictly less powerful than 2-FWL. RRWP is a natural extension of RWSE, which not only uses return probabilities $\mathbf{P}_{ii}^t$, but also uses non-diagonal elements $\mathbf{P}_{ij}^t$ as augmented edge features. However, it is still based on node-level random walk and can be completely simulated by 2-FWL. Combining our new results and the results from [51], we provide the following summary for the expressive power of node-level PE/SEs:

- RWSE is strictly less powerful than 2-FWL.
- SPDPE (shortest path distance) is strictly less powerful than 2-FWL.
- RDPE (resistance distance) is strictly less powerful than 2-FWL, but more powerful than SPDPE in distinguishing non-isomorphic distance-regular graphs.
- RRWP is strictly less powerful than 2-FWL, but it is strictly more powerful than SPDPE.
- HK (heat kernel) is strictly less powerful than 2-FWL.

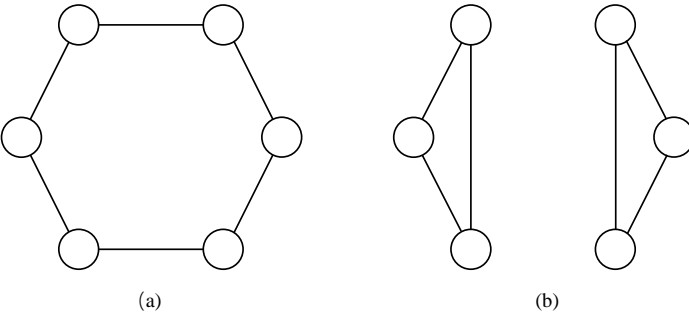

Figure 2: A pair of non-isomorphic graphs that are indistinguishable by 1-FWL. Graph (a) is a six-cycle, while graph (b) consists of two 3-cycles. Both 0-RWSE and Hodge-0 isospectra are able to distinguish these two graphs.

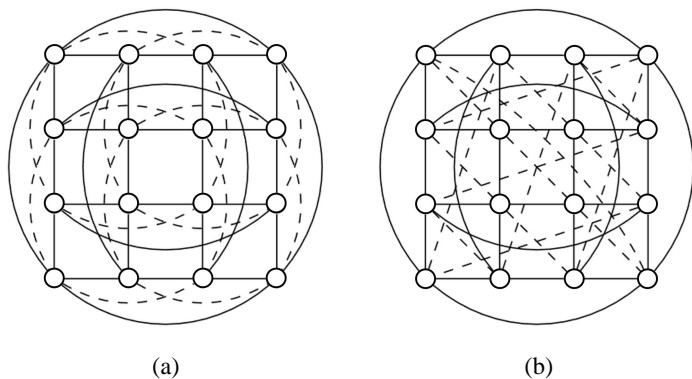

Figure 3: A pair of non-isomorphic graphs that are indistinguishable by 2-FWL. Graph (a) is the $4 \times 4$ Rook's graph, graph (b) is the Shrikhande's graph, both of them are strongly regular graphs parameterized by (16,6,2,2). Both 0-RWSE and Hodge-0 isospectra fail to distinguish these them, while full EdgeRWSE (1-RWSE) and Hodge-1 isospectra can distinguish them.

### C.1.2 Theoretical analysis of Hodge 0 spectra

As we will analyze Hodge 1-Laplacians in the following text in detail, Hodge 0-Laplacians that have been widely studied are not our main concentrations. In this section, we briefly summarize the properties, aiming to compare with Hodge 1-Laplacians.

**Theorem C.3.** *Hodge* 0*-isospectra is incomprable with* 1*-FWL and not more powerful than* 2*-FWL.*

*Proof.* Rattan and Seppelt [39] has already proved that Hodge 0-isospectra is upper-bounded by 2-FWL. Here we show that graphs in Figure 2 which are indistinguishable by 1-FWL can be distinguished by Hodge 0-isospectra. The characteristic polynomial for $\mathbf{L}_1$ of graph (a) is

$$\det\left(\lambda\mathbf{I} - \mathbf{L}_0(a)\right) = (\lambda - 4)(\lambda - 3)^2(\lambda - 1)^2\lambda \tag{42}$$

The characteristic polynomial for $\mathbf{L}_1$ of graph (b) is

$$\det\left(\lambda\mathbf{I} - \mathbf{L}_0(b)\right) = (\lambda - 8)^9(\lambda - 4)^6\lambda \tag{43}$$

Hence they are distinguishable by Hodge 0 eigenvalues.

However, the graph pair in Figure 4 that can be distinguished by 1-FWL cannot be distinguished by Hodge 0 eigenvalues, since the characteristic polynomial are both

$$\det\left(\lambda\mathbf{I} - \mathbf{L}_0(a)\right) = \det\left(\lambda\mathbf{I} - \mathbf{L}_0(b)\right) = \lambda(\lambda - 3)(\lambda - 1)^2(\lambda^3 - 9\lambda^2 + 21\lambda - 7) \tag{44}$$

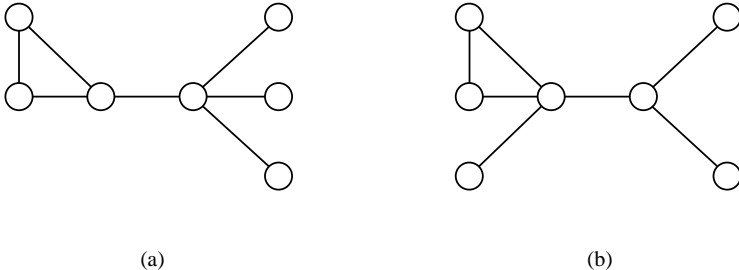



(a)                        (b)



Figure 4: A pair of non-isomorphic graphs that have isospectral Hodge $k$-Laplacians for all $k = 0, 1, 2, \ldots$, thus they are indistinguishable via eigenvalues of all Hodge $k$-Laplacians. However, they are 1-FWL distinguishable.

Therefore, there exist some 1-FWL distinguishable graph pairs that cannot be distinguished by Hodge-0 isospectra and vice versa. □

## C.2 Random walk on 1-simplices

### C.2.1 Theoretical analysis of 1-RWSE

We start by proving the expressive power of 1-RWSE (Full EdgeRWSE and its variations).

**Theorem C.4.** (First part of Theorem 5.1 in the main text). *Full EdgeRWSE can distinguish some non-isomorphic graphs that are indistinguishable by* 2*-FWL.*

*Proof.* We will show that a pair of strongly regular graphs parameterized by (16, 6, 2, 2), $4 \times 4$ Rook's graph and Shrikhande's graph (shown in Figure 3), are indistinguishable by 2-FWL but distinguishable by 1-RWSE. 2-FWL fails to distinguish any pair of strongly regular graphs with same parameters, which is a classical result, hence we refer readers to [8] for a complete proof and skip here.

Now we show that the full 1-RWSE is able to distinguish these two graphs. Only the 1-up random walk (i.e. walk towards those 1-up neighbors via shared 2-faces) needs to be considered, since the 1-down random walks on two graphs are identical (see Theorem C.5). For 1-up random walk, note that all edges in the same graph share the same status, thus we only need to analyze the random walk rooted at one arbitrary edge in two graphs, respectively. All edges in two graphs are contained in two different 2-simplices (triangles). However, the two 2-faces in (a) $4 \times 4$ Rook's graph belong to the same 3-simplex or 4-clique, but there are no 4-cliques in (b) Shrikhande's graph. Note that all 2-simplices in graph (a) are only connected to the 2-simplices that are in the same 3-simplex; consequently, an 1-up random walk on graph can only visit 4 different 2-simplices (which are all faces of the 3-simplex or 4-clique the source edge is in), and can only visit 6 different edges in the 4-clique including the source itself. But the 2-simplices in graph (b) are all connected, so the 1-up random walk has the probability of visiting all triangles and edges in the graph. Therefore, as time tends to infinity, the limit return probability of a 1-up random walk on $4 \times 4$ Rook's graph is

$$\lim_{t \to \infty} p^{\mathrm{R}}_{1,up}(t) = \frac{1}{6} \tag{45}$$

The limit return probability of a 1-up random walk on Shrikhande's graph is

$$\lim_{t \to \infty} p^{\mathrm{S}}_{1,up}(t) = \frac{1}{48} \tag{46}$$

Also, one can verify the difference on two walks via the return probabilities at the third step (or any other following steps, since the return probability on Rook's graph is always larger than that of Shrikhande's graph when $t \geq 3$):

$$p_{1,up}^{\text{R}}(3) = \frac{1}{8}, p_{1,up}^{\text{S}}(3) = \frac{1}{16} \tag{47}$$

Therefore, we have shown 1-up random walk can distinguish these two graphs, so can 1-random walk. Therefore, 1-RWSE can distinguish this pair indistinguishable from 2-FWL.

$\square$

Note that the 1-down RWSE is unable to distinguish $4 \times 4$ Rook's graph and Shrikhande's graph, since it does not consider any 2-simplicial faces. Similarly, 0-RWSE is also unable to distinguish them, which is consistent with our previous results. Actually, both 1-down RWSE and 0-RWSE are upper bounded by 2-FWL, see below.

**Theorem C.5.** (Second part of Theorem 5.1 in the main text.) 1-down RWSE is not more powerful than 2-FWL.

*Proof.* When the graph has no triangle, the edge level random walk matrix is

$$P_1 = B_1^T D_1^{-1} B_1 \tag{48}$$

while the node level random walk matrix is

$$P_0 = D_1^{-1} B_1 B_1^T \tag{49}$$

Therefore, $\forall l, k$, $tr(P_1^{kl}) = tr(P_0^{kl})$, and thus we can determine the multiset of $\{(P_1^l)_{[u,v],[u,v]} | [u,v] \in E\}$. $\square$

Now we give more discussion on random walks on 1-simplices. We start by further interpretation of the normalized Hodge 1-Laplacian and the edge-level random walk that corresponds. The transition matrix defined by $-\frac{1}{2}\tilde{\mathbf{L}}_1 \mathbf{V}^T = \mathbf{V}^T \hat{\mathbf{P}}$ (see Section 5.1 in the main text) can be interpreted as follows: $\hat{\mathbf{P}} = \frac{1}{2}\mathbf{P}_{down} + \frac{1}{2}\mathbf{P}_{up}$, where $\mathbf{P}_{down}$ and $\mathbf{P}_{up}$ are the transition matrices determined by a lower-adjacent and upper-adjacent random walk. Specifically, in each step with probability of $0.5$ each, we take a step toward either upper or lower adjacent edges. (1) If the step is taken towards the upper adjacent face (2-simplex or oriented triangle), there are two cases: (1a) the edge has an upper adjacent face, then the edge uniformly transits to an upper adjacent edge with different orientation relative to the shared face. (2a) the edge does not have upper adjacent faces, then the walk will keep the same orientation of this edge or change the orientation with equal probability $0.5$ each. (2) If the step is taken toward the lower adjacent face (0-simplex or node), the walk transits along or against the edge direction to the lower adjacent nodes with each probability $0.5$. Then from the selected node, the walk further transmits to the target edges connected with the node with probability proportional to the upper degrees or the weights of the target edges.

One can easily verify that the limit distribution of the directed 1-down random walk (case 1) is the uniform distribution at all edges, independent of the starting position. In comparison, the limit distribution of the undirected 1-down random walk (case 2) is proportional to the number of 1-down neighbors of the edges (i.e., the number of edges that share one node with the interested edge), thus providing no information except the number of neighboring edges. However, the initial steps of both two types of edge-level 1-down random walk are capable of providing rich information on the graph structure.

Here we further explain why simplified 1-down EdgeRWSE variations are appropriate in the cases where the SCs contain few 2-simplices (triangles). Recall that the full edge-level 1-random walk will have probability $0.5$ to go through the upper adjacent 2-faces, but stays at the same edge (with the same probabilities to keep or change direction) if the edge is not a face of the 2-complex. In this case, the full edge-level random walk described by $\hat{\mathbf{P}}$ will almost have probability $0.5$ to stay on the same edge (or the edge with an inverse direction if the graph is directed). This results in the walk having larger probability to stay still, or the walk becomes 'lazy' and provides less information about

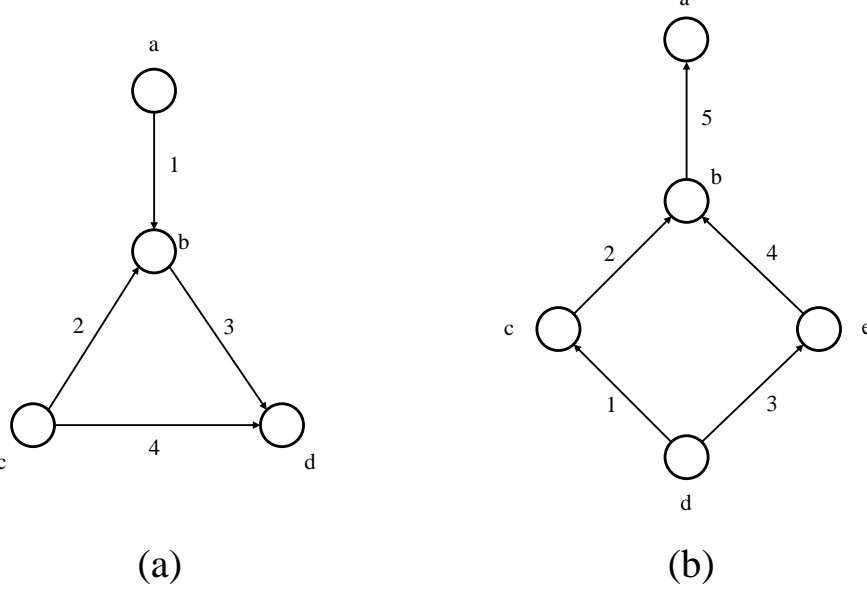

Figure 5: Two simplicial complexes with order 2 and 1, respectively.

surrounding structures. In comparison, the simplified walk variations neglect the upper adjacent 2-faces, thus having a greater probability of moving out of the source edge and exploring more about structure information. By reducing $\hat{\mathbf{P}}$ to $\mathbf{P}_{down}$, the simplified walk has more probability of moving away from the source, encouraging exploration of more information about the structure.

### C.2.2 Theoretical analysis of Hodge1Lap

We start analyzing the spectral properties of Hodge 1-Laplacians with proving the theorems in the main text. The following theorem discusses the relationship between spectra of Hodge 0-Laplacian $\mathbf{L}_0$ and Hodge 1-Laplacian $\mathbf{L}_1$.

**Theorem C.6.** (Theorem 5.2 in the main text.) *The number of non-zero eigenvalues of Hodge 1-Laplacian $\mathbf{L}_1$ is not less than the number of non-zero eigenvalues of Hodge 0-Laplacian $\mathbf{L}_0$.*

*Proof.* This is a direct conclusion from Equation (21) and Equation (22), which we have proved in Appendix A. □

A direct conclusion is that graph isomorphism based on Hodge 1-Laplacian isospectra (we only discuss the multisets of eigenvalues) is strictly more powerful than Hodge 0-Laplacian. The precondition here is that the two graphs have the same number of nodes and edges, or we can construct counter examples that two non-isomorphic graphs have the same $\mathbf{L}_1$ but have different numbers of nodes and different $\mathbf{L}_0$ (their $\mathbf{L}_0$ may have different dimensions and different multiplicities of zero eigenvalues if two graphs only differ from some extra isolated nodes). Under this condition, Equation (21) and Equation (22) state that if we can distinguish two graphs by their eigenvalues of Hodge 0-Laplacians, we can also do so by eigenvalues of Hodge 1-Laplacians. For the strictness, the graph pair in Figure 3 can be distinguished by Hodge 1 isospectra but not by Hodge 0 isospectra, see below.

**Theorem C.7.** (Theorem 5.3 in main text.) $\mathbf{L}_1$ *isospectra is incomparable with 1-FWL and 2-FWL.*

*Proof.* This theorem compares the expressive power of $\mathbf{L}_1$ isospectra with the traditional FWL hierarchy. First, we show that there are some non-isomorphic graph pairs distinguishable by 1-FWL and 2-FWL cannot be distinguished by $\mathbf{L}_1$ isospectra, see the example in Figure 4. 1-FWL can distinguish them for the following reasons. The first iteration of 1-FWL (equivalently, 1-WL) can hash the nodes of different degrees into different colors. Then in the second iteration, in graph (b) there is one node with degree 3 that owns two neighbors with degree 1. However, there is no such node in graph (a); thus, the two graphs have different multisets of colors after 2 iterations of 1-WL.

Therefore, both 1-FWL and 2-FWL can distinguish them. However, we do not distinguish them by the multisets of eigenvalues of $\mathbf{L}_1$. Both of their characteristic polynomials of $\mathbf{L}_1$ are:

$$\det\left(\lambda\mathbf{I} - \mathbf{L}_1(a)\right) = \det\left(\lambda\mathbf{I} - \mathbf{L}_1(b)\right) = (\lambda - 3)^2(\lambda - 1)^2(\lambda^3 - 9\lambda^2 + 21\lambda - 7) \quad (50)$$

Thus, they have identical multisets of eigenvalues of $\mathbf{L}_1$.

On the other hand, we show that $\mathbf{L}_1$ isospectra can distinguish strongly regular graphs in Figure 3, while 2-FWL and 1-FWL are known to fail on them. The characteristic polynomial for Hodge 1 Laplacian of $4 \times 4$ Rook's graph (which we denote as $\mathbf{L}_1(R)$) is

$$\det\left(\lambda\mathbf{I} - \mathbf{L}_1(R)\right) = (\lambda - 8)^9(\lambda - 4)^{30}\lambda^9 \quad (51)$$

The characteristic polynomial of Shrikhande's graph's Hodge 1-Laplacian $\mathbf{L}_1(S)$ is

$$\det\left(\lambda\mathbf{I} - \mathbf{L}_1(S)\right) = (\lambda - 8)^9(\lambda - 6)(\lambda - 3 - \sqrt{5})^6(\lambda - 4)^{15}(\lambda - 2)^9(\lambda - 3 + \sqrt{5})^6\lambda^2 \quad (52)$$

Obviously, their multisets of eigenvalues are different, hence Hodge 1-isospectra can distinguish some non-isomorphic graph pairs that even 2-FWL fail. $\qquad\square$

Now we turn our attention to the physical meanings and insights of the spectra of Hodge 1-Laplacians. Recall that zero eigenvalues and corresponding eigenvectors are associated with harmonic functions, and Hodge1Lap is able to figure out the cycles in the graph by projecting a unit vector into the subspace spanned by zero-eigenvalue eigenvectors (or equivalently, the kernel space of $\mathbf{L}_1$). Furthermore, the near-zero eigenvalues and their eigenvectors are associated with near-harmonic functions. The eigenvectors of large eigenvalues are non-homology generators for local structures such as clusters, stars, chains, and so on.

The following theorems state the properties of the Hodge 1-Laplacian and its harmonic space.

**Theorem C.8.** *In the Hilbert space, the $\mathbf{L}_1$ of a directed graph without multiple edges or self-loops is a $m \times m$ matrix, where $m$ is the number of directed edges. The elements of $\mathbf{L}_1$ satisfy: (1) the $i$-th diagonal element corresponding to the edge $e_i$ is $2 + |\delta_1(e_i)|$, where $|\delta_1(e_i)|$ is the number of 2-simplices $e_i$ is in; (2) element $\mathbf{L}_1[i, j] = 1$ if $e_i$ and $e_j$ are 1-down adjacent (share one node but not in the same 2-simplex) and two edges have the directions starting from the shared node; (3) $\mathbf{L}_1[i, j] = -1$ if $e_i$ and $e_j$ are 1-down adjacent but have different directions starting from the shared node; (4) $\mathbf{L}_1[i, j] = 0$ if $e_i$ and $e_j$ are 1-up adjacent (in the same 2-simplex).*

*Proof.* This can be directly verified by the definition of $\mathbf{L}_1 = \mathbf{B}_k^*\mathbf{B}_k + \mathbf{B}_{k+1}\mathbf{B}_{k+1}^*$. $\qquad\square$

We take Figure 5 (a) as an illustrative example of calculating $\mathbf{L}_1$. This is a simplicial complex of order $K = 2$. When writing matrices, we follow the order $a, b, c, d$ for 0-simplices (nodes) and the order $1, 2, 3, 4$ for 1-simplices (directed edges). The node-to-edge incidence matrix is

$$\mathbf{B}_1 = \begin{bmatrix} -1 & 0 & 0 & 0 \\ +1 & +1 & 0 & -1 \\ 0 & -1 & -1 & 0 \\ 0 & 0 & +1 & +1 \end{bmatrix} \quad (53)$$

Therefore the Hodge-0 Laplcian is (recall that $\mathbf{B}_0$ is not defined and is omitted in calculating $\mathbf{L}_0$)

$$\mathbf{L}_0 = \mathbf{B}_1\mathbf{B}_1^* = \begin{bmatrix} +1 & -1 & 0 & 0 \\ -1 & +3 & -1 & -1 \\ 0 & -1 & +2 & -1 \\ 0 & -1 & -1 & +2 \end{bmatrix} = \mathbf{D} - \mathbf{A} \quad (54)$$

where $\mathbf{D}$ is the diagonal matrix with node degrees as diagonal elements, and $\mathbf{A}$ is the adjacency matrix (in the normal sense). This is completely identical to the widely used graph Laplacian.

Now we come to compute $\mathbf{L}_1$. There is one 2-simplex in the simplicial complex, and we take the direction $c \to b \to d$ as the positive orientation of the 2-simplex, then

$$\mathbf{B}_2 = \begin{bmatrix} 0 \\ +1 \\ -1 \\ +1 \end{bmatrix} \tag{55}$$

and we have the Hodge-1 Laplacian

$$\mathbf{L}_1 = \mathbf{B}_1^* \mathbf{B}_1 + \mathbf{B}_2 \mathbf{B}_2^* = \begin{bmatrix} +2 & +1 & 0 & -1 \\ +1 & +2 & +1 & -1 \\ 0 & +1 & +2 & +1 \\ -1 & -1 & +1 & +2 \end{bmatrix} + \begin{bmatrix} 0 & 0 & 0 & 0 \\ 0 & +1 & -1 & +1 \\ 0 & -1 & +1 & -1 \\ 0 & +1 & -1 & +1 \end{bmatrix} = \begin{bmatrix} +2 & +1 & 0 & -1 \\ +1 & +3 & 0 & 0 \\ 0 & 0 & +3 & 0 \\ -1 & 0 & 0 & +3 \end{bmatrix}$$
$$\tag{56}$$

which is identical to the description in Theorem C.8. It can be seen that $\mathbf{B}_1^* \mathbf{B}_1$ contributes to (1), (2), (3) in Theorem C.8; $\mathbf{B}_2 \mathbf{B}_2^*$ contributes to (1) and (4) in Theorem C.8. In some sense, if $e_i$ and $e_j$ are both 1-down adjacent and 1-up adjacent, the elements $e_{ij}$ or $e_{ji}$ in $\mathbf{B}_1^* \mathbf{B}_1$ and $\mathbf{B}_2 \mathbf{B}_2^*$ cancel out each other, resulting $e_{ij} = e_{ji} = 0$ if $e_i$ and $e_j$ belong to the same 2-simplex.

Next, we declare some properties about the kernel space of $\mathbf{L}_1$ (i.e., the 1-cohomology group). Note that the following cycles are that of length larger than 4 since directed triangles are considered as 2-simplices, and in the sense of the undirected graph induced by the directed graph (i.e., we do not care about the directions of edges in the cycle, since any changes of direction are equivalent multiplying $-1$ on the original basis).

**Theorem C.9.** *For a directed graph without multiple edges, self-loops or 2-simplices that contains $c$ cycles, and there are no cycles that share edges, then a group of linearly independent and orthogonal basis of the kernel space of $\mathbf{L}_1$ is $\mathbf{u}_i, i = 1, \ldots, c$, where elements in $\mathbf{u}_i$ corresponding to the edges on the $i$-th cycle is $\pm \frac{1}{\sqrt{l(i)}}$ and $0$ otherwise, where $l(i)$ is the length of the $i$-th cycle.*

*Proof.* We have already stated that the direction of edges only affect the sign of corresponding element in eigenvectors, WLOG, we suppose that all edges in the $i$-th cycle follow the anticlockwise direction. Then all nonzero elements on the same basis $\mathbf{u}_i$ have identical signs. We are going to prove this.

Obviously, $\mathbf{u}_i, i = 1, \ldots, c$ are linearly independent and orthogonal, since the cycles do not share edges, the nonzero elements in $\mathbf{u}_i$ do not overlap consequently. Therefore, we only need to prove that $\mathbf{L}_1 \mathbf{u}_i = 0$. (1) For the $j$-th row where edge $e(j)$ is not in the $i$-th cycle, there are two cases: (1a) $e(j)$ is not connected with any edges in the $i$-th cycle, then obviously the inner product of $j$-th row of $\mathbf{L}_1$ and $\mathbf{u}_i$ is zero, since all nonzero elements in the $j$-th row of $\mathbf{L}_1$ corresponds to zero elements in $\mathbf{u}_i$; (1b) $e(j)$ is connected with some edges in the cycle, then the only situation is that $e(j)$ connects with two edges $e(j_1), e(j_2)$, where the shared node of these three edges is the source node of $e(j_1)$ and is the target node of $e(j_2)$, and the $j$-th row of $\mathbf{L}_1[j]$ will have $-1$ in element $j_1$, $+1$ in element $j_2$. Therefore, the inner product $\mathbf{L}_1[j]^T \cdot \mathbf{u}_i = 0 + 2 \times 0 + (-1) \times \frac{1}{\sqrt{l(i)}} + 1 \times \frac{1}{\sqrt{l(i)}} = 0$. (2) If $e(j)$ is in the cycle $i$, then similar to (1b), the non-zero index in both $\mathbf{L}_1[j]^T$ and $\mathbf{u}_i$ is the index of two neighboring nodes of $e(j)$, hence $\mathbf{L}_1[j]^T \cdot \mathbf{u}_i = 0 + 2 \times \frac{1}{\sqrt{l(i)}} + 2 \times (-1) \times \frac{1}{\sqrt{l(i)}} = 0$. Putting all the pieces together, we have proved $\mathbf{L}_1 \mathbf{u}_i = 0$. $\square$

Figure 5 (b) represents an example that satisfies the above conditions in the theorem. For illustration, we calculate its $\mathbf{L}_1$

$$\mathbf{L}_1 = \begin{bmatrix} +2 & -1 & -1 & 0 & 0 \\ -1 & +2 & 0 & +1 & -1 \\ -1 & 0 & +2 & +1 & 0 \\ 0 & +1 & +1 & +2 & -1 \\ 0 & -1 & 0 & -1 & +2 \end{bmatrix} \tag{57}$$

It is straightforward to verify that the eigenvector corresponding to the zero eigenvalue is

$$\mathbf{u}_0 = \pm \frac{1}{2} [1, 1, 1, -1, 0]^\top \tag{58}$$

The elements corresponding to edges in a cycle (edge $1, 2, 3, 4$) are non-zero, while other elements are strictly zero. Therefore, Hodge1Lap is capable of detecting cycles through the eigenvectors in $\ker(\mathbf{L}_1)$.

**Corollary C.10.** *Suppose that the above conditions hold, denote* $\mathbf{U} = [\mathbf{u}_1, \dots, \mathbf{u}_c]$. *Then for a vector* $\mathbf{e}$ *of length* $m$ *and all elements* $1$, *then projection* $\left|\mathbf{U}\mathbf{U}^T\right|\mathbf{e}$ *has* $j$-th *element* $1$ *if* $e(j)$ *is in a cycle and* $0$ *otherwise.* $\left|\left|\right.\right.$ *indicates taking the absolute values element-wise.*

*Proof.* This is a direct conclusion of Theorem C.9. Note that $|\mathbf{U}\mathbf{U}^T|$ has element $\left|\mathbf{U}\mathbf{U}^T\right|[i, j] = \frac{1}{l(c_{ij})}$ if both edges $i$ and $j$ are in the cycle $c_{ij}$, and $0$ otherwise. Hence the projection of $\mathbf{e}$ in the kernel space has element $1$ if the corresponding edge is in a cycle and $0$ otherwise, which means that this projection can efficiently mark out cycles. Note that the projection matrix is actually independent of the choice of basis; hence the above conclusion is universal. For the basis matrix $\mathbf{V}$ where the columns are independent but not orthogonal, the projection matrix should be calculated as $\mathbf{V}(\mathbf{V}^T\mathbf{V})^{-1}\mathbf{V}^T$. $\qquad\square$

**Corollary C.11.** *Suppose that the above conditions hold, except that two cycles* $c_1, c_2$ *share one edge* $e(j)$. *Then in every eigenvector* $\mathbf{u}$ *in the kernel space of* $\mathbf{L}_1$, $\mathbf{u}_j = \gamma_1 - \gamma_2$, *where* $\gamma_1, \gamma_2$ *are the edge flow and are positive following the anticlockwise direction,*

$$|\mathbf{u}_i(i \neq j)| = \begin{cases} \gamma_1, & e(i) \in c_1 \\ \gamma_2, & e(i) \in c_2 \\ 0, & e(i) \notin c_1, e(i) \notin c_2 \end{cases} \tag{59}$$

*Further,* $(\mathbf{U}\mathbf{U}^T\mathbf{e})_j = 0$.

*Proof.* This is also straightforward to verify, since the basis in Theorem C.9 still holds for shared edges among cycles. However, the two edge flows in the cycle definitely have opposite directions in the same edge, as we assign anticlockwise as the positive direction for the flows. Consequently, the flow in $e(j)$ is the difference of two edge flows. Moreover, for the projection without operation of taking element-wise absolute value in the projection matrix, the projection of two cycles in the $j$-th element of $\mathbf{e}$ will cancel out. $\qquad\square$

There are also some additional interesting conclusions, for example, if $e(j)$ is in cycle $c$ and in a 2-simplex, then the edge flow in $e(j)$ will be $\frac{2}{3}\gamma$, where $\gamma$ is the original edge flow in the cycle, and the edge flow in the other two edges of the 2-simplicial will be $\frac{1}{3}\gamma$. Moreover, the eigenvalue 1 of $\mathbf{L}_1$ will mark out the following substructure: node $a, c$ are both only connected to $b$, while $b$ is also connected to a fourth node $d$. All these interesting results can be verified through simple algebra, and we will not list all of them due to the limited space.

Finally, despite the satisfying theoretical properties of Hodge1Lap if we use the projection method (Hodge1Lap$_{\text{proj}}$ in the main text), we experimentally find that a naive summation over the absolute values of eigenvectors (referred as Hodge1Lap$_{\text{abs}}$) works fine for a number of real-world tasks, although this method is sign-invariant but not basis-invariant.

We also provide more discussion on the cycle detecting ability of Hodge1Lap. As explained above, in Hodge1Lap$_{\text{proj}}$, projecting the constant vector onto the kernel space of $\mathbf{L}_1$ will result in the element $j$ equal to 1 if the edge $j$ is in a cycle, and 0 otherwise. However, it is unable to distinguish different cycles or to indicate which cycle an edge belongs to. In comparison, Hodge1Lap$_{\text{abs}}$ can distinguish and detect different cycles, which we explain as follows. In the main text, this implementation is described as $\sum_i \phi(|\mathbf{v}_i|)$, where $||$ indicates taking element-wise absolute values, and $\mathbf{v}_i$ refers to the interested eigenvectors (e.g. orthogonal basis of kernel space of $\mathbf{L}_1$, in case that we are discussing detecting cycles). Note that here $\mathbf{v}_i$ is different from $\mathbf{u}_i$ described in Theorem C.9; instead, they are arbitrary random eigenvectors, and we aim to use randomness to distinguish different cycles. For simplicity, we first do not consider overlapping or 2-simplexes; then every eigenvector $\mathbf{v}_i$ corresponding to zero eigenvalues has the following properties: (i) the elements corresponding to edges that are not in any cycles are zero; (ii) the elements corresponding to edges from the same cycles have identical absolute values, while those of edges from different cycles (almost surely) have

different absolute values. These properties can be easily verified, since any $\mathbf{v}_i$ can be represented as a linear combination of $\mathbf{u}_i$ described in Theorem C.9. Therefore, $\sum_i MLP(|\mathbf{v}_i|)$ obtains different values for edges from different cycles, which is why we distinguish and detect them. In practical implementations, we further apply a random projection technique to make Hodge1Lap more robust to the choice of basis $\mathbf{v}_i$: $\sum_j MLP(\sum_i \alpha_{ij}|\mathbf{v}_i|)$, where $\alpha_{ij} \sim \mathcal{N}(0,1)$ are random variables, and $j = 1, \ldots, J$ where $J$ is the maximum number of projections. Through this design, the model can explicitly learn different cycles and implicitly maintain the basis invariance.

## C.3 Random walk on higher order and inter-order simplices

### C.3.1 Properties of random walk on higher order and inter-order simplices

Analogously to 0-simplices and 1-simplices, we can use random walk on any $k$-simplices to build expressive $k$-RWSE in order to facilitate graph and simplicial learning. Meanwhile, Hodge $k$-Laplacians are closely connected with random walk of $k$ order, whose spectra, including their eigenvalues and eigenvectors, are applicable in methods such as PE or spectral convolution learning.

Here we prove some crucial properties of Hodge Laplacians $\mathbf{L}_k$.

**Lemma C.12.**
$$\mathbf{L}_k^r = (\mathbf{L}_{k,up} + \mathbf{L}_{k,down})^r = \mathbf{L}_{k,up}^r + \mathbf{L}_{k,down}^r \tag{60}$$

*Proof.* We prove this by induction. For $r = 1$, the equation obviously holds.

Suppose $\mathbf{L}_k^r = \mathbf{L}_{k,up}^r + \mathbf{L}_{k,down}^r$, then for its $r+1$-th power, we have

$$\mathbf{L}_k^{r+1} = (\mathbf{L}_{k,up}^r + \mathbf{L}_{k,down}^r)(\mathbf{L}_{k,up} + \mathbf{L}_{k,down}) \tag{61}$$
$$= \mathbf{L}_{k,up}^{r+1} + \mathbf{L}_{k,down}^{r+1} + \mathbf{L}_{k,up}^r \cdot \mathbf{L}_{k,down} + \mathbf{L}_{k,down}^r \cdot \mathbf{L}_{k,up} \tag{62}$$
$$= \mathbf{L}_{k,up}^{r+1} + \mathbf{L}_{k,down}^{r+1} + (\delta_k^* \delta_k)^r \delta_{k-1} \delta_{k-1}^* + (\delta_{k-1} \delta_{k-1}^*)^r \delta_k^* \delta_k \tag{63}$$
$$= \mathbf{L}_{k,up}^{r+1} + \mathbf{L}_{k,down}^{r+1} \tag{64}$$

The last equation holds because $\delta_k \delta_{k-1} = 0$ and $\delta_{k-1}^* \delta_k^* = 0$ always hold for all $k \geq 1$. $\qquad\square$

Using the above property, we can prove that the power of the inter-order adjacent matrix (defined in Section 6.2 in the main text) satisfies the following property.

**Theorem C.13.** (Equation 7 in main text).

$$\mathcal{A}_K^r = \begin{bmatrix} p_r(\mathbf{L}_0) & q_{r-1}(\mathbf{L}_{0,up})\delta_0^* & & & \\ q_{r-1}(\mathbf{L}_{1,down})\delta_0 & p_r(\mathbf{L}_1) & q_{r-1}(\mathbf{L}_{1,up})\delta_1^* & & \\ & \cdots & & \cdots & & \\ & & \cdots & \cdots & & \cdots \\ & & & q_{r-1}(\mathbf{L}_{K,down})\delta_{K-1} & p_r'(\mathbf{L}_K, \mathbf{L}_{K,down}) \end{bmatrix} \tag{65}$$

*where $p_r(\cdot)$ and $q_r(\cdot)$ are polynomials with maximum order $r$, except the last row is incomplete without $\mathbf{L}_{K,up}$. Note that we replace $\mathbf{B}_{k+1}^*$ in main text with coboundary operators $\delta_k$ for universality.*

*Proof.* We still prove by induction. For $r = 1$, the equation obviously holds with $p_1(x) = x, q_0 = 1$. Suppose the equation holds for $r$, then the $r+1$ power satisfies the following.

(1) The block at first row and first column concerning $\mathbf{L}_0$, note that $\mathbf{L}_0 = \mathbf{L}_{0,up} = \delta_0^* \delta_0$, we have

$$\mathcal{A}_K^{r+1}[1,1] = p_r(\mathbf{L}_0)\mathbf{L}_0 + q_{r-1}(\mathbf{L}_{0,up})\delta_0^* \delta_0 = \left(p_r(\mathbf{L}_0) + q_{r-1}(\mathbf{L}_0)\right)\mathbf{L}_0 = p_{r+1}(\mathbf{L}_0) \tag{66}$$

(2) The $k$th diagonal block ($1 < k < K$):

$$\mathcal{A}_K^{r+1}[k,k] = q_{r-1}(\mathbf{L}_{k,down})\delta_{k-1}\delta_{k-1}^* + p_r(\mathbf{L}_k)\mathbf{L}_k + q_{r-1}(\mathbf{L}_{k,up})\delta_k^* \delta_k \tag{67}$$
$$= \left(p_r(\mathbf{L}_k) + q_{r-1}(\mathbf{L}_k)\right)\mathbf{L}_k \tag{68}$$
$$= p_{r+1}(\mathbf{L}_k) \tag{69}$$

where we use $q_{r-1}(\mathbf{L}_{k,up}) + q_{r-1}(\mathbf{L}_{k,down}) = q_{r-1}(\mathbf{L}_k)$ according to Lemma C.12. Further, we have $p_{r+1}(x) = (p_r(x) + q_{r-1}(x)) \cdot x$.

(3) The block at $k$-th row and $k+1$-th column:

$$\mathcal{A}_K^{r+1}[k, k+1] = 0 + p_r(\mathbf{L}_k)\delta_k^* + q_{r-1}(\mathbf{L}_{k,up})\delta_k^*(\delta_k\delta_k^* + \delta_{k+1}^*\delta_{k+1}) + 0 \tag{70}$$

$$= p_r(\delta_{k-1}\delta_{k-1}^*)\delta_k^* + p_r(\delta_k^*\delta_k)\delta_k^* + q_{r-1}(\mathbf{L}_{k,up})\mathbf{L}_{k,up} \tag{71}$$

$$= \Big(p_r(\mathbf{L}_{k,up}) + q_{r-1}(\mathbf{L}_{k,up})\mathbf{L}_{k,up}\Big)\delta_k^* \tag{72}$$

$$= q_r(\mathbf{L}_{k,up})\delta_k^* \tag{73}$$

Hence we have $q_r(x) = p_r(x) + q_{r-1}(x) \cdot x$.

Further, the block at $k+1$-th and $k$-th column is the adjoint of that:

$$\mathcal{A}_K^{r+1}[k+1, k] = \Big(q_r(\mathbf{L}_{k,up})\delta_k^*\Big)^* = \delta_k \cdot q_r(\delta_k^*\delta_k) = q_r(\mathbf{L}_{k+1,down})\delta_k \tag{74}$$

(4) The block at $k$-th row and $k+2$-th column:

$$\mathcal{A}_K^{r+1}[k, k+1] = 0 + 0 + q_{r-1}(\mathbf{L}_{k,up})\delta_k^*\delta_{k+1}^* = 0 \tag{75}$$

Therefore, the block in $k+2$-th row and $k$-th column is also $0$.

(5) The other blocks are obviously zero.

Combining all these pieces together, we prove the theorem.

$\square$

The above equation states that simplices with difference of order larger than one cannot directly exchange information even after infinite rounds, but they can affect each other through the coefficients in $p_r$ and $q_{r-1}$ in the blocks on the offset $\pm 1$-diagonal blocks.

It's noticable that a number of previous works such as [8] can be reformatted and unified by $\mathcal{A}_K$. Additionally, we can make use of $\mathcal{A}_K^r$ to build random walk based positional encoding for all simplices in $K$-dimensional simplicial complex that contains more information than random walks within the same order simplices.

In addition, analogously to [7], we can introduce any form of discrete topological structures, e.g., cellular complex, as expanded complex cells. Our methods can also be naturally extended to discrete domains without orientations, e.g. hypergraphs. In particular, we can define random walk on these discrete topological structures, which can greatly facilitate graph learning by incorporating structures of higher order other than simplicial complexes. Our methods can be easily generalized to these structures, for example, we provide the results of random walk on cellular complexes in Table 1, which treats cycles as 2-cellular complexes and greatly improve the performance of the base model. More implementation details can be found in Appendix E. However, the theoretical analysis including expressive power in distinguishing non-isomorphic graphs of these new methods are nontrivial due to the flexibility in definition of complex cells, which is a future direction worth exploring.

### C.3.2 Hodge Laplacians spectra and graph isomorphism

In the spectral domain, a necessary but insufficient condition for two graphs to be isomorphic is that their Hodge $k$-Laplacians are isospectral (having the same eigenvalues) for all $k \geq 0$. As we have discussed before, the Hodge 0-isopectra is incomparable to 1-FWL and is not more powerful than 2-FWL. Hodge 1-isospectra is incomparable with 1-FWL and 2-FWL. Note that Hodge $k+1$-isospectra is not necessarily more powerful than Hodge $k$-isospectra, except that for $k=0$ the conclusion holds since $\mathbf{L}_{0,down} = 0$. However, we can always get a more powerful algorithm by increasing the highest order of Hodge Laplacians while maintaining all the Hodge Laplacians of lower orders.

In addition to making use of eigenvalues, we can build more powerful GNNs based on the spectra of Hodge $k$-Laplacians. We address that we can universally approximate permutation equivariant and basis invariant functions (moreover, $k$-form) defined on $k$-simplicial complexes if we use Expressive-BasisNets [32]. Universality is guaranteed by the following decomposition theorem.

**Theorem C.14.** (Theorem 4 in [32].)*Theorem 4 (Decomposition Theorem). Let $\mathcal{X}_1, \ldots, \mathcal{X}_k$ be topological spaces and let $G_i$ be a topological group that continuously acts on $\mathcal{X}_i$ for each $i$. We assume that the mild topological conditions on $\mathcal{X}_i$ and $G_i$ hold. Assume that there is a topological embedding $\psi_i : \mathcal{X}_i/Gi \to \mathbb{R}^{a_i}$ of each quotient space into a Euclidean space $\mathbb{R}^{a_i}$ for some dimension $a_i$. Then, for any continuous function $f : \mathcal{X} = \mathcal{X}_1 \times \cdots \times \mathcal{X}_k \to \mathbb{R}^{d_{out}}$ that is invariant to the action of $G = G1 \times \cdots \times G_k$, there exist continuous functions $\phi_i : \mathcal{X}_i \to \mathbb{R}^{a_i}$ and a continuous function $\rho : \mathcal{Z} \subseteq \mathbb{R}^a \to \mathbb{R}^{d_{out}}$, where $a = \sum_i a_i$ such that $f(v_1, \ldots, v_k) = \rho(\phi_1(v_1), \ldots, \phi_k(v_k))$.*

The proof of Theorem C.14 is completed in [32]. For mild topological conditions, we only need $G_i$ to be a topological group that continuously acts on $\mathcal{X}_i$ for each $i$, and to know that there exists a topological embedding of each quotient space into some Euclidean space. As a matter of fact, the continuous group action is a very mild assumption which holds for any finite or compact matrix group. For a finite simplicial complex, the eigenvalues and eigenvectors of Hodge-Laplacians are finite; thus, the compactness naturally holds. Furthermore, using Hodge-Laplacians normalization in [24], the maximum eigenvalue of $\mathbf{L}_k$ satisfies $\lambda_{max} \leq k + 2$. Finally, letting $\mathcal{X}_i$ be the cochain $\mathcal{C}^i$ in the above theorem, we see that the permutation equivariant and basis-invariant function as well as the $k$-form defined on the simplicial complex can be universally approximated, for instance, by Expressive-BasisNet.

Although we can recover the spectra of $\mathbf{L}_k, k = 0, \ldots, K$ (or spectra convolution on simplicial complexes) via Expressive-BasisNet, it is actually impractically expensive. On the other hand, we can use spectral information, such as the eigenvalues and eigenvectors of $\mathbf{L}_k$ to facilitate learning via simple networks.

## D    Random walk message passing

In our main text, we mainly discuss how to facilitate graph and simplicial learning through designing PE and SE based on random walk on simplicial complexes. However, it is remarkable that random walk on simplicial complexes can give different insights and inspirations in designing powerful GNNs and simplicial networks in addition to PE and SE. In this section, we propose a novel random walk message passing (RWMP) mechanism, which introduces node distance metrics and simulates a weighted random walk at node level by dropping edges according to distances between neighboring nodes. As revealed in the experiments shown in the main text, RWMP is able to improve the performance of the base models and achieves SOTA performance in the Zinc dataset [17].

### D.1    Connections between random walk, subgraph sampling and edge dropping

There have been a great number of relevant works on improving GNNs with subgraph sampling and edge dropping. Subgraph GNNs are a huge family of GNN variations that encode a set of subgraphs instead of the original graph. Some subgraph GNNs are more powerful than MPNN or 1-WL such as [5], while Frasca et al. [22] upper bound node-based subgraph GNNs by 3-WL. Despite the improvement in expressivity and performances, a number of subgraph GNNs suffer from extensive computation complexity due to the exponential growth of the subgraph amount. Therefore, various subgraph sampling methods have been proposed to improve the scalability of subgraph GNNs. For example, $K$-hop GNN [21] extracts a $K$-hop subgraph for each root node, and Zeng et al. [49], Toenshoff et al. [43] propose to sample subgraphs via a random walk started from a root node. Due to its internal association with the diffusion process, random walk is a powerful tool to design subgraph sampling methods. Meanwhile, these subgraph GNNs based on sampling in a random walk fashion can be interpreted as implicitly encoding structure information obtained by random walk via the following procedure: sampling subgraphs (sampling according to probability transition matrix), running GNN on the subgraphs (encoding structure captured by a random walk), and aggregating information among subgraphs (taking expectation).

Dropping edge is another family of methods that introduce randomness into graph learning. DropEdge [40] is a widely adopted technique that randomly drops edges during training to improve the generalizability of GNNs. Unlike DropEdge, DropGNN [37] randomly drops edges in both the training and inference stages. DropGNN is able to partly improve expressive power, but it has to gather several rounds of inference results to obtain the final representation. In the context of representation learning for linear features, the representations obtained by randomly dropping edges layer-wise are unbiased estimators of full message passing if we consider normalization. In comparison, the

representations obtained by subgraph sampling are biased estimators of full message passing. The layer-wise drop edge can be viewed as an intermediate product between full message passing and subgraph sampling GNNs, which (1) obtains a representation in every forward pass like original message passing and does not have to run several forward passes on subgraphs in parallel; (2) has randomness like subgraph sampling methods and has the probability to encode subgraph structures. We leave further discussion and more theoretical analysis for future work.

### D.2 Random walk message passing: weighted random walk based on node-level distance metric

As we discussed above, both the subgraph sampling and edge-dropping methods can be associated with the simple node-level random walk on unweighted graphs. Through introducing randomness, they can better encode structure information and improve expressive power of GNNs. However, these methods use uniform sampling and uniform dropping strategies associated with unweighted random walk, and hence they are only able to learn pure structure information but no feature information.

To address this limitation, we propose the Random Walk Message Passing mechanism (RWMP), which can jointly learn feature and structure information by introducing distance metrics of nodes (or node features). Briefly speaking, RWMP drops edges in a layerwise fashion, and the probability of dropping edge $e = (u, v)$ is a function of the distance between nodes $u$ and $v$: $P_{\mathrm{drop}}(e) = f(d(u, v))$, where $f$ is the designed function, and $d(u, v)$ is the (feature) distance between $u$ and $v$ in the defined metric. For example, if we use cosine similarity between node features as the distance metric, a possible drop probability is:

$$P_{\mathrm{drop}}(e) = \frac{1}{2}(1 - \frac{||\mathbf{u} \cdot \mathbf{v}||}{||\mathbf{u}|| \cdot ||\mathbf{v}||}) \tag{76}$$

where $\mathbf{u}$ and $\mathbf{v}$ denote the node feature vector of $u$ and $v$. The above metric states that the probability of message passing between two nodes is linear with their cosine similarity in node features. The more they are different, the less likely they are to exchange information with each other. Intuitively, this can partly address the problem of oversmoothing, since it can reserve heterogeneity by reducing message passing times within dissimilar nodes. RWMP also encourages message passing within similar nodes, which is beneficial for discovering local clusters (it is equivalent to sample the subgraphs containing local structures of similar nodes with larger probabilities).

RWMP is a novel framework that can jointly learn structure and feature information by simulating a weighted random walk based on a distance metric. Appropriate designs of distance metrics are able to greatly improve the performance of RWMP, which is applicable to any message-passing based GNN variations. RWMP can be regarded as an intermediate scheme between deterministic message passing variations like GAT [45] and subgraph sampling-based methods. However, on the one hand, RWMP is different from GAT in the way that RWMP explicitly introduces randomness via dropping edges instead of taking expectation through the attention weights, and the stochastic process forces RWMP to explore more about local structures while relieving oversmoothing. On the other hand, RWMP greatly reduces the computation costs on subgraph GNNs - the latter often fail to run on large graph benchmarks, while preserving the ability to explore more on certain structures.

Combining RWMP with edge-level PE/SE, a GINE [25] model is able to achieve highly competitive performance in the Zinc dataset. More experimental details are listed in Appendix E.

## E Experiments

### E.1 Datasets description

In Section 7 in the main text and Appendix E.2, we conduct extensive experiments on various datasets, confirming the effectiveness of our methods. Among these datasets, Zinc, MNIST, and CIFAR10 are from Benchmarking GNN [17], ogbg-molhiv and ogbg-molpcba are from Open Graph Benchmark [26], while PCQM-Contact, Peptides-func and Peptides-struct are from Long-range Graph Benfchmark [19].

Table 3: Experiments on synthetic datasets (ACC ↑). The backbone model is GINE, which is not more expressive than 1-WL.

| PE/SE | EXP | SR25 |
|---|---|---|
| None | 50 | 6.67 |
| Hodge1Lap(eigenvalues) | 100 | 100 |
| EdgeRWSE(full) | 100 | 100 |

Table 4: Experiments on two datasets from benchmarking GNN [17]. Highlighted are the first, second, **third** test results.

| model | MNIST (Accuracy ↑) | CIFAR10 (Accuracy ↑) |
|---|---|---|
| GCN [28] | $90.705 \pm 0.218$ | $55.710 \pm 0.381$ |
| GIN [46] | $96.485 \pm 0.252$ | $55.255 \pm 1.527$ |
| GAT [45] | $95.535 \pm 0.205$ | $64.223 \pm 0.455$ |
| GatedGCN [10] | $97.340 \pm 0.143$ | $67.312 \pm 0.311$ |
| PNA [15] | $97.94 \pm 0.12$ | $70.35 \pm 0.63$ |
| DGN [4] | - | $72.838 \pm 0.417$ |
| CRaWl [43] | $97.944 \pm 0.050$ | $69.013 \pm 0.259$ |
| GIN-AK+ [53] | - | $72.19 \pm 0.13$ |
| EGT [27] | $98.173 \pm 0.087$ | $68.702 \pm 0.409$ |
| GPS [38] | $98.051 \pm 0.126$ | $\mathbf{72.298 \pm 0.356}$ |
| GatedGCN+EdgeRWSE | $\mathbf{98.069 \pm 0.115}$ | $70.260 \pm 0.341$ |
| GPS+EdgeRWSE | $98.245 \pm 0.070$ | $72.417 \pm 0.221$ |

**Zinc.** Zinc-12k is a subset of Zinc-250k, which consists of 12000 molecular graphs from the ZINC database of commercially available chemical compounds. The task is to perform graph-level molecular property regression (on constrained solubility logP). These molecular graphs are between 9 and 37 nodes large. We follow the common predefined 10K/1K/1K train/validation/test split.

**MNIST and CIFAR10.** These two datasets both contain directed graphs, and are derived from like-named image classification datasets. Both of them are 10-class classification tasks, and we follow the standard dataset splits as the original image classification datasets, i.e., 55K/5K/10K for MNIST and 45K/5K/10K for CIFAR10 of train/validation/test graphs, respectively.

**ogbg-molhiv and ogbg-molpcba.** They are both molecular property prediction datasets, using a common node (atom) and edge (bond) featurization to represent chemophysical properties. In detail, the prediction task of ogbg-molhiv is a binary classification of the fitness of a molecule to inhibit HIV replication, which is measured by AUROC. The task of ogbg-molpcba is a 128-task binary classification, evaluated by average precision.

**PCQM-Contact.** It is a dataset derived from PCQM4Mv2 and the corresponding 3D molecular structures. The task is a binary link prediction which is evaluated by the Mean Reciprocal Rank (MRR).

**Peptides-func and Peptides-struct.** These two datasets both consist of atomic graphs of peptides. The task for Peptides-func is a multi-label graph classification into 10 nonexclusive peptide functional classes measured by average precision. The task for Peptides-struct is graph regression of 11 3D-structural properties of the peptides measured by MAE.

### E.2  Full experiment results

While SOTA and highly competitive results on Zinc, ogbg-molhiv and ogbg-molpcba are shown in the main text, we further discuss the performance of our methods on other datasets here, including synthetic datasets and real-world datasets. For real-world datasets including BenchmarkingGNN [17] and Long Range Graph Benchmark (LRGB) [19], we follow the experimental settings of [38] and

Table 5: Experiments on three datasets from long-range graph benchmarks (LRGB) [19]. Highlighted are the first, second, **third** test results.

| model | Peptides-func (AP ↑) | Peptides-struct (MAE ↓) | PCQM-Contact (MRR ↑) |
|---|---|---|---|
| GCN | $0.5930 \pm 0.0023$ | $0.3496 \pm 0.0013$ | $0.3234 \pm 0.0006$ |
| GINE | $0.5498 \pm 0.0079$ | $0.3547 \pm 0.0045$ | $0.3180 \pm 0.0027$ |
| GatedGCN | $0.5864 \pm 0.0077$ | $0.3420 \pm 0.0013$ | $0.3218 \pm 0.0011$ |
| Transformer+LapPE | $0.6326 \pm 0.0126$ | $0.2529 \pm 0.0016$ | $0.3174 \pm 0.0020$ |
| SAN [30]+LapPE | $0.6384 \pm 0.0121$ | $0.2683 \pm 0.0043$ | $\mathbf{0.3350 \pm 0.0003}$ |
| SAN+RWSE | $0.6439 \pm 0.0075$ | $0.2545 \pm 0.0012$ | $\mathbf{0.3350 \pm 0.0003}$ |
| GPS | $\mathbf{0.6535 \pm 0.0041}$ | $0.2500 \pm 0.0005$ | $0.3337 \pm 0.0006$ |
| GatedGCN+EdgeRWSE | $0.6002 \pm 0.0048$ | $0.2679 \pm 0.0015$ | $0.3342 \pm 0.0008$ |
| GatedGCN+Hodge1Lap | $0.5926 \pm 0.0059$ | $0.2632 \pm 0.0008$ | $0.3336 \pm 0.0004$ |
| GPS+EdgeRWSE | $0.6625 \pm 0.0042$ | $0.2501 \pm 0.0012$ | $0.3408 \pm 0.0003$ |
| GPS+Hodge1Lap | $0.6584 \pm 0.0033$ | $\mathbf{0.2505 \pm 0.0014}$ | $0.3407 \pm 0.0004$ |

do not perform a hyperparameter search, since our main goal is to verify that our Hodge1Lap and EdgeRWSE can benefit the arbitrary base model.

**Experiments on synthetic datasets.** To verify the theoretical expressive power of Hodge1Lap and EdgeRWSE, we carried out experiments on two classical synthetic datasets: EXP and SR25. EXP [1] contains 600 pairs of non-isomorphic graphs that 1-WL and 2-WL fail to distinguish. SR25 [3] contains 15 non-isomorphic strongly regular graphs (i.e., 105 non-isomorphic pairs) that 3-WL fails to distinguish. An accuracy of 50% on EXP and 6.67% on SR25 suggests the model fails to distinguish any non-isomorphic graphs in the dataset. The results are shown in Table 3. The 1-WL equivalent GINE back-end model can fully distinguish EXP and SR25 when augmented by Hodge1Lap and Full-EdgeRWSE, indicating that both methods can distinguish non-isomorphic graph pairs that 3-WL (and 2-FWL) fails, which is consistent with our theoretical analysis.

**Experiments on MNIST and CIFAR10.** Since the graphs are all directed, only the directed version of EdgeRWSE is applicable since a random walk can only go along the edge, while Hodge1Lap is also applicable if we follow the strict directions of edges and simplices. Here, we evaluate the effectiveness of EdgeRWSE acting on GatedGCN and GPS, see Table 4. With EdgeRWSE, the GPS model (which consists of a GatedGCN and a Transformer in each layer) achieves the best on MNIST and second best on CIFAR10. It is remarkable that a simple GatedGCN is greatly enhanced by EdgeRWSE, achieving highly competitive performance on both datasets. This verifies the effectiveness of EdgeRWSE in models involving edge features.

**Experiments on LRGB.** The results are shown in Table 5; the baseline results are adopted from [38]. On both Peptides-func and PCQM-Contact, GPS model with EdgeRWSE (we use undirected version) and Hodge1Lap (we use projection method) achieve the best and second best performance, respectively. On Peptides-struct, the improvement of edge-level PE/SE is not significant for GPS, but is obvious for the GatedGCN base model. Admittedly, as pointed out by [44], the baseline models need some reassessing including hyper-parameter searching, and there are more SOTA results. However, we emphasize that our results are obtained without hyperparameter searching - our main goal is to verify the enhancement brought by EdgeRWSE and Hodge1Lap to the baseline models. SOTA methods can always be facilitated with our PE/SE to achieve better performance.

**Trajectory prediction.** Since our methods originate from simplicial complexes, we also verify the effectiveness of our Hodge1Lap on simplicial data. We adopt the trajectory prediction task (edge flow classification) in [8]. The edge flows are represented as signals on oriented simplicial complexes. We use the same synthetic dataset of trajectories on the simplicial complex in [8], where each triangle is treated as a 2-simplex. There are two holes in the complex, and all trajectories pass one of the holes, thus giving rise to two different classes to be distinguished. Due to the presence of the two holes, the trajectories of the two classes approximately correspond to orthogonal directions in the space of harmonic eigenfunctions of the $\mathbf{L}_1$ Hodge-Laplacian [41]. The dataset contains

Table 6: Trajectory classification accuracy on synthetic flow dataset with simplicial networks.

| model | Train accuracy | Test accuracy |
|---|---|---|
| MPSN [8] $L_0$-inv | $88.2 \pm 5.1$ | $85.3 \pm 5.8$ |
| MPSN - Id | $88.0 \pm 3.1$ | $82.6 \pm 3.0$ |
| MPSN - Tanh | $97.9 \pm 0.7$ | $95.2 \pm 1.8$ |
| MPSN $L_0$-inv + Hodge1Lap | $98.4 \pm 0.7$ | $99.8 \pm 0.4$ |
| MPSN - Id + Hodge1Lap | $99.5 \pm 0.2$ | $99.3 \pm 0.3$ |
| MPSN - Tanh + Hodge1Lap | $\mathbf{100.0 \pm 0.0}$ | $\mathbf{99.9 \pm 0.1}$ |

1000 train trajectories and 200 test trajectories. Following [8], to make the task more challenging for non-orientation-invariant models, all training complexes use the same orientation for the edges, while the test trajectories use random orientations. We adopt orientation-invariant message-passing simplicial networks (MPSN) [8] with orientation equivariant layers as the base models.

Note that since we aim to distinguish two orthogonal directions in the harmonic kernel space of $\mathbf{L}_1$ Hodge-Laplacian, we do not use the projection implementation of Hodge1Lap; instead, we use the simple implementation $\mathrm{Hodge1Lap}_{\mathrm{sim}}$ (described in the main text) which embeds two eigenvectors respectively with a shared MLP. Theoretically, we can distinguish the two classes of flows with the help of eigenvectors since they correspond to two distinct cycles (i.e., two orthogonal directions in the kernel space). As shown in Table 6, all variants of MPSN achieve better performance (almost perfect test accuracy) when they are augmented with our Hodge1Lap. This is consistent with our theory, which experimentally verified the capability of Hodge1Lap to distinguish cycles.

## E.3 Experiment details

**Dataset split and repetitiveness.** In all experiments, the datasets follow the common train/validation/test split as we stated in datasets description. The results of the baseline models are reported from [38]. For our results, we report the test results according to the best validation results. All experiments are run under 5 different random seeds, with mean and variance reported.

**Hyperparameters.** To verify that our methods are capable of improving the performance of the base models, all hyperparameters including training configuration and model hyperparameters are set the same as in [38]. For the edge PE/SE, we keep the embedding dimensions the same as the node PE/SE in GPS models. Hence, our experimental results strongly confirmed that our methods are beneficial for both naive and complex base models.

## E.4 Implementation details

In this subsection, we provide more implementation details of our methods and more discussion on the corresponding variants. Our code is based on GPS [38], which enable us to integrate our methods into a comprehensive graph learning framework.

**EdgeRWSE.** In real-world datasets, we use both directed 1-down RWSE and undirected 1-down RWSE in our experiments, except in MNIST and CIFAR10 where only a directed walk is appropriate on the directed graphs. We also integrate variance of each row in the transition matrix corresponding to the target edge in each step by an MLP. Generally, directed and undirected versions reveal similar performance, and we report the performance of undirected EdgeRWSE if not specific.

For Full EdgeRWSE, we verify its theoretical expressive power through experiments on synthetic datasets; see Table 3. A GINE enhanced with Full EdgeRWSE achieves $100\%$ performance on the synthetic datasets EXP and SR25. To fully distinguish EXP and SR25, the model needs to be more expressive than 1-WL and 2-FWL, respectively. This shows that Full-EdgeRWSE is (partly) more powerful than 2-FWL (3-WL), which is consistent with our theory. In comparison, 1-down EdgeRWSE that does not consider 2-simplices fails ($0\%$) to distinguish strongly regular graphs in SR25. However, we experimentally find that the performance of Full-EdgeRWSE on zinc dataset is similar to 1-down EdgeRWSE with a GINE base model. This is because 2-simplices (or triangles)

are actually rare in molecules, so Full-EdgeRWSE makes no significant difference than 1-down EdgeRWSE.

We also implement the interorder random walk among $0, 1, 2$-simplices. The random walk probability matrix is the normalized $\mathcal{A}_2$ in our main text. We compute $\mathcal{A}_2, \ldots, \mathcal{A}_2^T$ ($T = 20$ for Zinc) and embed the diagonal elements for both node and edge features. The performance of GINE augmented with Inter-RWSE is significantly better than pure GINE but is slightly weaker than simultaneously applying NodeRWSE and EdgeRWSE (which are separately computed without inter-order communications). We attribute it to: (i) lack of tuning hyper-parameters such as $T$, and (ii) the return probabilities are much less in inter-order random walk, as there are totally $n + m$ possible targets, instead of $n$ for 0-random walk and $m$ for 1-random walk. This makes it harder for the model to learn meaningful structure information and may require larger $T$.

**CellularRWSE**     The concept of CellularRWSE is explained in Appendix C.3, where we treat the edges as 1-cells and extract all the rings (cycles) as 2-cells. In this CellularRWSE, a random walk is able to transit from one source edge to (i) another target edge sharing one node, or (ii) another target edge that is in the same 2-cell as the source edge. The performance is slightly better than EdgeRWSE (see Table 1), which may be due to the ability to distinguish higher-order structures (rings and cycles).

**Hodge1Lap.**     We already include a detailed discussion in the main text and in Appendix C.2.2 on two different implementations: projection-based methods and sign-invariant methods. As we have shown, the projection method $\text{Hodge1Lap}_{\text{proj}}$ is both sign and basis invariant, and we consider the subspace spanned by eigenvectors of zero eigenvalues. The sign-invariant method $\text{Hodge1Lap}_{\text{abs}}$ simply takes absolute values of eigenvectors element-wise, and uses a MLP to encode eigenvectors and eigenvalues. We additionally apply a MLP to learn eigenvalue-eigenvector pairs of Hodge-1 Laplacians in our Hodge1Lap implementation. In practice, we adopt the combination of these two methods; if not specific, we report the results of the combined variant.

**RWMP.**     We use the cosine similarities of the node features as our distance metric. The drop probability follows Equation (76). In each layer, we use a normal GINE layer without RWMP and a GINE layer with RWMP and add their features as the final representation of the layer. The GINE [25] model with Hodge1Lap and RWMP achieves highly competitive performance on Zinc, outperforming GPS [38] and Specformer [6].

**Other implementation details.**     Some other implementation details and discussions are summarized below.

- For SSWL+, we add edge PE/SE features embedded by a unique linear projection to the original edge features in every layer.
- For GRIT, the PE is embedded by a linear layer and then directly added to the edge features. For virtual edges, the PE values are padded as zeros.
- Our method is can be applied to higher-order GNNs, including simplicial networks [8, 7] and 2-$\delta$-FWL equivalent classes [36], while others (SPD, RD) cannot. Our method is also more sparse.
- Our method can be applied to any base model without increasing much computation cost and can be applied together with other levels of PE/SE.
- There are more than one way to integrate PE/SE into the models, while we choose the most simple ones: we treat PE/SE as initialization of edge features by concatenating or adding them to the original edge features; they can also be used as in graphormer (as attetion biases), or any convolution/attention computation in simplicial networks.

### E.5   Complexity and applicability analysis

Theoretically, the complexity to generate Hodge1Lap and EdgeRWSE are both $O(m^2)$, where $m$ is the number of edges. For small or sparse graphs, they share similar complexity with node-level PE such as LapPE and RWSE. For example, the average generation times on Zinc computed by RTX3090 are: RWSE (23s), EdgeRWSE (32s), Hodge1Lap (28s). Even for extremely large graphs with millions of nodes, we can handle by sampling subgraphs as we are only interested in the local

structure information for some PE/SE. For example, for a $K$-step EdgeRWSE, we can sample a $K$-hop subgraph rooted at the target edge and continue to process, without having to deal with large matrix multiplication. It is noticeable that we only have to **preprocess once** for every dataset, so the precompute time is negligible compared to experiment time.

For the computation complexity in the forward pass of models, since all our PE/SE are embedded by light-weight MLPs, the extra computation cost is completely ignorable compared with the whole model. In summary, our methods are able to significantly enhance model performance with extremely small additional computation cost.

Regarding applicability, on the base model side, our Hodge1Lap and EdgeRWSE are universally applicable to any base model as long as edge features are considered. Our methods are orthogonal to other PE/SE and can be used together. SOTA methods can always be facilitated with our methods to achieve better performance. On the data type side, as we have emphasized, our methods are applicable to (both directed and undirected) graphs, simplicial complexes, and other discrete topological data structures, e.g. cellular complexes.

