# OpenReview forum: "Facilitating Graph Neural Networks with Random Walk on Simplicial Complexes"
_NeurIPS.cc/2023/Conference — NeurIPS 2023 poster_

### Official Review · Reviewer_j6x1 · 2023-07-02

**Soundness:** 3 good
**Presentation:** 1 poor
**Contribution:** 3 good
**Rating:** 5
**Confidence:** 2

**Summary:**

The authors generalize node-level random walk positional and structural encoding to edge-level (1-simplex) and higher-order simplex random walks. They focus on the edge-level for both theoretical and empirical results. Their approach is motivated primarily through theory, and then expanded empirically to the algorithm level. In addition, the authors prove important results about their new algorithm, allowing their architecture to be strictly more expressive than 1-FWL. Empirically, their results outperform their selected baselines.

**Strengths:**

The paper's strengths lie in their novel approach to generalizing node-level random walks to edge-level and beyond. Although no empirical evidence is provided for the higher-levels, we see convincing empirical evidence that edge-level features contribute to the overall performance of the model. To the best of my knowledge, this is one of the first times that Hodge Laplacians have been used in GNN contexts, allowing the authors to apply their algorithms to practical examples.

**Weaknesses:**

The paper's weaknesses lie in the lack of extremely convincing empirical results and the lack of clarity in the paper.

Empirically, the authors' new algorithms generally perform slightly better than their baselines; however, in around half the datasets provided, there exist other models which perform better than their algorithms. If possible, it would be more convincing to build their new algorithms on the current state-of-the-art models. Experiments on synthetic datasets that require models to be strictly more expressive than 1-FWL would also be helpful here (presumably, models with the proposed algorithms would perform drastically better).

The paper is unclear at times. Although the theory is interesting by itself, the importance of the results that were derived is not clear, and the algorithms are not explicitly defined anywhere. Explicit "algorithm" sections describing EdgeRWSE and Hodge1Lap would add much to the paper. Finally, the authors mention a "novel Random Walk Message Passing scheme" for the first time in the experiments section, with no prior explanation anywhere else in the paper. It seems a bit isolated from the rest of the paper; I would rather it be described in the body or in a separate manuscript entirely, instead of trying to fit into an already dense paper.

There also exist many grammar and syntactical mistakes scattered throughout the paper (examples: line 150 typo, 173 grammar, 263 typo).

**Questions:**

1. Are the new proposed approaches equivalent to RWSE and other approaches in terms of theoretical time complexity and practical compute time, memory, and parameters?
2. Why were GINE, GPS, and GATEDGCN chosen as base models to augment?

**Limitations:**

The authors do not include many limitations in their work, but based on my understanding of the work, I believe that all limitations are implicitly addressed.

I believe there are no potential negative societal impacts of their work.

---

> ### Author Rebuttal · Authors · 2023-08-10
>
> Thank you for your review points. We address your questions as follows.
>
> Weakness
>
> 1. Applying to SOTA baselines: We have applied our method to SOTA baselines, including GRIT[5] and Exphormer[6]. We also improve the performance by improving the implementation and hyper-paremeters of Hodge1Lap and EdgeRWSE. Please refer to general response for details.
>
> [5] Graph Inductive Biases in Transformers without Message Passing. ICML 2023.
>
> [6] Exphormer: Sparse Transformers for Graphs - Shirzad, Velingker, Venkatachalam, Sutherland, Sinop. ICML 2023.
>
> 2. Experiments on synthetic datasets: We have conducted experiments on synthetic datasets, including EXP and SR25, two widely adopted synthetic datasets to measure theoretical expressivity. Both Hodge1Lap and Full-EdgeRWSE are able to achieve perfect performance on both two datasets, while other PE/SE fail. The results are consistent with our theoretical analysis that Hodge1Lap and Full-EdgeRWSE can distinguish some non-isomorphic graph pairs that $1$-WL and $3$-WL ($2$-FWL) fail. It's noticable that none of SPDPE, RDPE or RWSE can distinguish SR25 since they are strictly bounded by $2$-FWL in expressive power.
>
> 3. Importance of our theories: We believe our theories are great contribution to both GNN theoretical expressive power and application communities. The results in Section 3 of supplementary (covering Section 4 - Section 6 in main text) are the first to reveal properties of PE and SE based on random walk on higher-order simplices as well as Hodge-$k$ Laplacians. We also make connections with existing $k$-WL hierarchies, showing that previous PE/SE are limited in their expressive power, while Hodge1Lap and EdgeRWSE can provably improve the expressivity of GNNs. Besides unifying existing methods including PE/SE and simplicial networks, we also propose novel Hodge1Lap/EdgeRWSE designs, along with a blue-print for designing higher-order and inter-order random walk based PE/SE methods.
>
> 4. Writing and explicit algorithms: we do describe and discuss different possible implementation of Hodge1Lap and EdgeRWSE in our paper. We believe that equations (3) and (5) in main text is clear enough to express the design idea while enjoying great flexibility. There is no algorithm section in our original text, since our methods enjoy extremely flexible design space. For clarification, the general procedure is to compute positional/structure encoding statistics (e.g. eigenvalues and eigenvectors of $\mathbf L_1$), and use some models (e.g. MLPs) to encode the statistics following certain rules (e.g. projection to the null space). However, we will improve writing add some paragraphs describing implementation details in the revision.
>
> 5. RWMP: We do not mention RWMP in the main text due to limited space. However, this can be naturally integrated into any (weighted) random walk, as RWMP introduces a distance metric based on features. Therefore, it has close connections with the rest of our papers under the framework of random walk. The idea is also novel and explores a way to design more flexible GNN architectures. We will reorganize the paper in the final version and briefly mention it in main text.
>
> Questions
>
> 1. The computation depends on number of edges. Theoretically, the complexity for generating Hodge1Lap and EdgeRWSE are both $O(m^2)$, where $m$ is the number of edges. For common graphs, they basically share the same compute time and memories with node-level ones like RWSE. For example, the average generation time on Zinc computed by RTX3090 are: RWSE ($23$s), EdgeRWSE ($32$s), Hodge1Lap ($28$s). The number of parameters depends on detailed architectures, e.g. a linear layer or a 3-layer MLP as used in most of our experiments. Generally speaking, our method is extremely parameter-efficient but largely increase the performance.
>
> 2. We generally follow GPS. GINE and GatedGCN are message-passing based GNN, while GPS integrates graph transformers. We show that our method can benefit both these types of graph machine learning models. GPS[4] also report that these architectures can be improved by PE/SE. Moreover, in our new experiments we apply our methods to over eight base models, verifying the universality of our methods.
>
> [4] Recipe for a general, powerful, scalable graph transformer.

---

> > ### Comment · Reviewer_j6x1 · 2023-08-18
> >
> > I thank the authors for their detailed responses. I appreciate the new results and am interested to see how the reorganization of the paper would look like. I will maintain my score for now.

---

> > > ### Author Response · Authors · 2023-08-19
> > >
> > > Thank you for your review points and your reply. We are glad that you are satisfied with our responses and the new results. For the final version of our paper, we will reorganize the paper and improve our writing to make it clearer, and include all these new results as well as more details to enrich the paper. We believe that our paper will be a huge contribution.

---

### Official Review · Reviewer_kVcw · 2023-07-06

**Soundness:** 3 good
**Presentation:** 3 good
**Contribution:** 3 good
**Rating:** 6
**Confidence:** 3

**Summary:**

A major issue with graphs is the absence of positional information on nodes, which decreases the representation power of GNNs to distinguish e.g., isomorphic nodes. An approach to tackle this issue is to introduce Positional Encoding (PE)  and Structural Encoding (SE) of nodes and inject it with node features. Most PE and SE techniques are based on node-level (0-simplices ) and edge-level (1-simplices ) random walk features. This paper analyzed the theoretical expressivity of node-level and edge-level random walks. The authors also introduce EdgeRWSE and Hodge1Lap,  new edge-level random walk-based features that make GNNs more powerful.  The authors analyzed theoretically the expressivity of these methods. They also introduced and generalized the study to random walks on higher-order simplices. The proposed approaches were tested empirically on many datasets and achieved competitive results with state-of-the-art methods.

**Strengths:**

- The paper is well-written and organized. I appreciated sections about 0-simplices,1-simplices and high-order simplices.

- It introduces a reasonable way to facilitate the expressivity of GNNs through positional and structural encodings.

- The paper studied and analyzed the theoretical expressivity of node-level, edge-level random walks.

- The paper generalized the random walks to high-order simplices and analyzed their expressivity.

**Weaknesses:**

The main weaknesses of the paper have to do with the empirical analysis.

-   The combination of GNNs with EDGERWSE and HODGE1LAP has been tested only for GPS and GINE architectures. To validate the assumption that ‘EDGERWSE and Holde1LAp improve the performance of GNNs’, the combination should be tested for other GNN architectures as well, e.g., GIN, PNA, etc. Especially since the proposed positional and structural encoding is applicable to any standard GNN.

- The experiments have been performed for graph-level tasks only (mainly graph classification and graph regression tasks). What is the performance of such approaches on node classification tasks?

- The comparison with previous random walk-based PE and SE is missing, i.e., methods in papers [31,1,30,19]. EDGERWSE and Holde1LAp should be compared with the PE and SE baselines using the same GNN architecture.

- The time complexity for generating EDGERWSE and Holde1LAp is missing.

**Questions:**

In the weaknesses part above, I have included most of my questions to the authors. One more question follows below.

- From Tables 1, 2 in the main paper and Tables 1, 2 in the supplementary material,  I couldn't see a huge improvement in the performance when using EDGERWSE and Holde1LAp methods. Could you please further elaborate on this point?

**Limitations:**

Some of the limitations of the proposed methodology have been discussed in the paper.

---

> ### Author Rebuttal · Authors · 2023-08-10
>
> Thank you for your valuable reviews. We address the weakness and your questions as follows.
>
> 1. Test on more GNN architectures: In our new experiments, we apply our Hodge1Lap and EdgeRWSE to a variety of different GNN models, see Table 1 and general response for more details. For all base models, we verify that our methods are capable of improving empirical performances.
>
> 2. Node classification tasks: Generally, node classification tasks do not need so much structure information, and papers in the relevant field of GNN expressive power usually pay more attention to graph-level tasks. However, to show that our method is also beneficial for node-level tasks, we add experiments on Brazil-airports datasets [8]. We use GAT as our baseline model and evaluate the effectiveness of EdgeRWSE. Since the node classification is closely related to local structures, number of layers of GAT is set to 3 as in [8], and number of walk steps of EdgeRWSE is set to 10. We report mean test accuracy (with validation) over 20 seeds. By treating EdgeRWSE as additional edge features, the performance improves from $71.24\pm 3.89$ to $75.83\pm 2.61$, exceeding the best performance reported in [8] ($75.37\pm 3.25$ for DEA-GNN-SPD). We are still working on more node-level tasks and will report the progress.
>
> [8] Distance Encoding: Design Provably More Powerful
> Neural Networks for Graph Representation Learning.
>
> 3. Comparison with other random walk based PE/SE: Theoretically we have compared the expressive power of our methods and these random walk based PE/SE. Experimentally, we use GRIT as the common back-end model, and compare our methods with RWSE, SPDPE, RDPE and RRWP (reported in Table 1). We observe that Hodge1Lap and EdgeRWSE alone can improve the performance of backbone model, but slightly weaker than other PE/SE methods. However, we want to emphasize: (i) different models and tasks may benefit from different PE/SE, which is also observed in [4]; (ii) while SPDPE and RDPE are applied to $n\times n$ node-pairs (only applicable to dense connectivity), our Hodge1Lap and EdgeRWSE only have values for real-edges, which is more sparse and more generalizable (directly applicable for real sparse edges, and applicable to dense connectivy via padding; (iii) our methods can be integrated with existing methods (e.g. node-level PEs), resulting highly competitive results.
>
> [4] Recipe for a general, powerful, scalable graph transformer.
>
> 4. Computation complexity: Theoretically, the complexity for generating Hodge1Lap and EdgeRWSE are both $O(m^2)$, where $m$ is the number of edges. For small or sparse graphs, they share similar complexity with node-level PE such as LapPE and RWSE. For example, the average generation time on Zinc computed by RTX3090 are: RWSE ($23$s), EdgeRWSE ($32$s), Hodge1Lap ($28$s). Even for extremely large graphs with millions of nodes, we can handle by sampling subgraphs as we are only interested in the local structure information for some PE/SE. For example, for a $K$-step EdgeRWSE, we can sample a $K$-hop subgraph rooted at the target edge and continue to process, without having to deal with large matrix multiplication. It's noticeable that we only have to preprocess once for every dataset, so the pre-compute time is negligible compared with experiment time.
>
> 5. Performance: We actually achieve several SOTA performance on Zinc and LRGB benchmarks. We also adopt some novel embedding methods (see general response for implementation details) and improve the performances. Again, we address that our methods are applicable to all kinds of GNN variations, but the performance gain depends on the base models and the tasks.

---

> ### Author Response · Authors · 2023-08-21
> **We are looking forward to your reply**
>
> We thank reviewer for the review points, and we have answered your questions in detail while making every effort to improve the experiments. We are looking forward to your reply to see if we have properly addressed your concerns. Your response will definitely help to make a more reasonable decision on our paper.

---

> > ### Comment · Reviewer_kVcw · 2023-08-22
> > **Reviewer response**
> >
> > I would like to thank the authors for their detailed responses. Discussing the main issues raised in my review, namely experiments on node classification, comparison to other random walk-based PE/SE, and complexity analysis, definitely help to better understand the functioning of the proposed methodology. Considering the comments of the other reviewers, I have accordingly modified my score.

---

> > > ### Author Response · Authors · 2023-08-22
> > > **Thank you**
> > >
> > > We sincerely appreciate your review suggestions and your valuable acknowledgment.

---

### Official Review · Reviewer_r8fz · 2023-07-06

**Soundness:** 4 excellent
**Presentation:** 2 fair
**Contribution:** 3 good
**Rating:** 7
**Confidence:** 3

**Summary:**

The paper studies random walks over simplicial complexes to construct novel positional and structure encoding methods for improving graph neural networks.
In particular, the authors propose two strategies to construct edge embeddings, namely EdgeRWSE (via random walks on the edges) and Hodge1Lap (via spectral analysis of the Hodge 1 Laplacian).
Finally, these constructions are theoretically generalized to higher order simplices for generic simplicial complexes.


**Strengths:**

The paper is clearly written.
The contributions are relevant and the ideas proposed seem novel.
In particular, I liked the idea of using the Hodge Laplacian's kernel to enrich the GNN with some information about the graph's cycles.


**Weaknesses:**

Section 6 generalises the method to higher order simplexes but it doesn't seem necessary in the main paper, which instead discusses only normal graphs.
This generalisation is not used in the experimental section either.
I feel like Section 6 could be safely moved in the supplementary material to keep the paper more focused on the main applications and contributions, but I am open to discussions about this.
This could free some space to include the details about the RWMP method, which is currently missing from the main paper.



**Questions:**


lines 248-249: the use of the terms sign-invariant and basis-invariant seems confusing since I don't see a group action. It seems more natural to describe this issue in terms of "uniquess only up to a sign / change of basis". I would at least mention this fact more explicitly to motivate the choice of words sign/basis-invariant.



lines 268-272: while the kernel space of $L_1$ can count all cycles (as a vector space, it has a dimension per cycle), detecting individual cycles is - as far as I know - a more complex problem which boils down to finding particular sparse bases for the first (co)homology group (i.e. the kernel of $L_1$).
The projection of the constant vector $\boldsymbol{e}$ in eq. 5 only spans a 1 dimensional subspace of the (co)homology group so it can not detect all cycles.
For this reason, I find the claim that Hodge1Lap can extract cycles in a natural way requires some further comments and clarification.


line 289: could you provide some intuition about why the lifting is done this way? For example, why is the lifting done to a 2-times larger space rather than a $(k+1)!$ times larger one (one copy for every permutation of the nodes in a simplex)?



**Limitations:**

There is not explicit discussion of the limitations of the method.

---

> ### Author Rebuttal · Authors · 2023-08-09
>
> We thank the review for acknowledging our work and the useful advice. In the new revision of our paper, we will introduce RWMP in the main text. However, we believe that Section 6 is a necessary part. This section generalize our theories, showing that our method can be extended to any higher-order structures. We have also implemented some higher-order and inter-order PE/SE instances in our new experiments to make it more contributive.
>
> Now we answer your questions as follows.
>
> 1. Sign/basis invariance: To our knowledge, sign and basis invariant are actually widely considered in graph learning field, for example [19] gives a thorough discussion. Here sign and basis refer to the ones of eigenspaces instead of group action. However, it's indeed easier for common readers to understand if we describe it as "invariant to the choice of sign and basis of eigenvectors".
>
> [19] Sign and Basis Invariant Networks for Spectral Graph Representation Learning - Derek Lim and Joshua Robinson and Lingxiao Zhao and Tess E. Smidt and Suvrit Sra and Haggai Maron and Stefanie Jegelka.
>
> 2. Ability to detect cycles: Besides counting cycles, the eigenvectors in the kernel space of $\mathbf L_1$ can indeed **detect individual cycles**, i.e. the basis for first cohomology group. We describe in detail how the cycles are counted and detected in **Section 3.2.2 of supplementary (Theorem 3.9-3.11)**. It's remarkable that in eq. (5), $P_{proj,i}$ is the projection matrix to a subspace with dimension $m_i$ instead of one-dimension, see eq. (4). If we want to detect cycles, $\lambda_i=0$, then $m_i$ is the multiplicity of zero eigenvalues, which is also the first-order Betti number $\beta_1$ (also the number of cycles). Hence, this $m_i=\beta_1$-dimensional subspace is able to detect all $m_i=\beta_1$ cycles. Please inform if you need any further clarificaions.
>
> 3. Explanation of lifting: This is because any higher-order k-simplex (k>0) only have 2 possible orientations instead of $(k+1)!$ (which is a fact that is often confused). Generally, the orientation is defined according ${\rm sgn}(\sigma)$ in eq. (1) of supplementary, where $\sigma$ is the element in the symmetric group $\mathcal G_{k+1}$. While there are $(k+1)!$ different permutations $\sigma$, there are only two directions depending on ${\rm sgn}(\sigma)$, i.e. whether it is an even or odd permutation. All even permutations are equivalent in their direction, so do all odd permutations. For example, for a triangle $k=2$, there are obviously only two directions instead of $(k+1)!=6$. This applies to all higher-order simplices. Since the lifting is only for enumerating possible directions, 2-times larger space is enough rather than $(k+1)!$ times larger one.

---

> > ### Comment · Reviewer_r8fz · 2023-08-15
> >
> > Thanks for the detailed answer.
> >
> > Regarding the ability to detect cycles, my comment regarding the one-dimensional subspace referred to the fact you only project the constant vector $\textbf{e}$. As a result, the output of the linear operation $P_{proj, 0} \textbf{e} = UU^T \textbf{e}$ "probes" only a one-dimensional subspace of the kernel of $L_1$.
> >
> > From Corollary 3.10-11, I understand that the scalar $[UU^T \textbf{e}]_j$ contains some information about the cycles the node $j$ belongs to.
> > For example, *if no cycles overlap*, $[UU^T \textbf{e}]_j$ tells whether the node $j$ belongs to a cycle or not.
> >
> > However, it seems to me this provides no information about which cycles each node belongs to.
> > For this reason, it seems the model is only able to detect whether the graph contains cycles or not, but it is not able to distinguish different cycles and, therefore, detect them.
> > This observation seems to also fit the intuition that the feature $UU^T \textbf{e}$ only catches the information contained in a 1-dimensional subspace, while detecting individual cycles requires the whole kernel of $L_1$.
> >
> > Moreover, when cycles overlap or the graph contains 2-simplexes, this information is further "diluted" over multiple nodes (see Corollary 3.11 and following discussion): do I understand correctly that only nodes belonging to tree-like leaf subgraphs will have zero values?
> > While this construction can be useful in graphs where cycles are clearly distinguished and non-overlapping, it seems less powerful when generic graphs are considered (this seems related to the fact that a cycle basis is rarely unique so the statement "a node belongs to a cycle" might not be well defined).
> >
> > Is my understanding correct?

---

> > > ### Author Response · Authors · 2023-08-15
> > > **Further clarifications on Hodge1Lap**
> > >
> > > Thank you for your response. Your understanding is basically right, but here are some points we want to further explain.
> > >
> > > - Hodge1Lap is an edge-level PE, which provide information about edges instead of nodes. Thus, every word 'node' in your previous comment should be replaced with 'edge'.
> > >
> > > - You are right that projecting the constant vector to the kernel space of $\mathbf L_1$ will result in element $j$ equals to 1 if edge $j$ is in a cycle, and 0 otherwise. It's unable to distinguish different cycles, nor indicating which cycle an edge belongs to. However, besides the projection method, we proposed another implementation variation of Hodge1Lap (namely ABS), see line 345 of main text and line 360-362 of supplementary. This variation can distinguish and detect different cycles, which we explain as follows. In the original text, this implementation can be described as $\sum_i MLP(|v_i|)$, where $||$ indicates taking element-wise absolute values, and $v_i$ refers to the interested eigenvectors (e.g. orthogonal basis of kernel space of $\mathbf L_1$, in case that we are discussing detecting cycles). For simlicity, we first do not consider overlapping or 2-simplexes, then every eigenvector $u_i$ corresponding to zero eigenvalues has the following properties: (i) the elements corresponding to edges that are not in any cycles are zero; (ii) the elements corresponding to edges from the same cycles have identical absolute values, while those of edges from different cycles (almost surely) have different absolute values. These properties can be easily verified, as any $v_i$ can be represented as a linear combination of $u_i$ described in Theorem 3.9. Therefore, $\sum_i MLP(|v_i|)$ obtains different values for edges from different cycles, hence distinguishing and detecting them. In practical implementations, we further apply a random projection technique to make Hodge1Lap more robust to the choice of basis $v_i$: $\sum_j MLP(\sum_i \alpha_{ij}|v_i|)$, where $\alpha_{ij}\sim \mathcal N(0,1)$ are random variables, and $j=1,\dots,J$ where $J$ is the maximum number of projection. Through this design, the model can explicitly learn different cycles and implicitly maintain basis invariance.
> > >
> > > - For cases where there are overlap or higher order simplexes, the computed PEs are indeed different from cases without them. However, the information is not 'diluted', as long as the model is able to learn the difference. As for 'nodes belonging to tree-like leaf subgraphs will have zero value', we guess you are actually referring to the shared edge between two cycles. And we can also mitigate this zero value of shared edge via computing $|\mathbf U||\mathbf U|^T \mathbf e$ (in which the shared edge of two cycles will have a larger value) instead of $\mathbf U\mathbf U^T \mathbf e$.
> > >
> > > In summary, the projection method indeed can only indicate whether an edge is in a cycle, but not which cycle it belongs to. However, this is still beneficial for cases where cycles are not dense, e.g. in most molecular graphs. Moreover, the ABS implementation of Hodge1Lap and the random projection technique are fully capable of distinguishing and detecting different cycles, even in cases where cycles are dense or overlapped. This is extremely beneficial for generic graphs, proving the effectiveness of Hodge1Lap.
> > >
> > > Hope that we have addressed your questions. Please let us know if you need more further clarification.

---

> > > > ### Comment · Reviewer_r8fz · 2023-08-16
> > > >
> > > > Thanks for the quick and detailed answer!
> > > >
> > > > I would encourage to include this discussion in the paper, since it seems very helpful to understand the capacity and the limits of the proposed approach.
> > > >
> > > > > Thus, every word 'node' in your previous comment should be replaced with 'edge'.
> > > >
> > > > That is true, sorry for my mistake!
> > > >
> > > > > You are right that projecting the constant vector to the kernel space of  will result in element  equals to 1 if edge  is in a cycle, and 0 otherwise. It's unable to distinguish different cycles, nor indicating which cycle an edge belongs to.
> > > >
> > > > Thanks for clarifying this! I'd again encourage the authors to mention this in the discussion in page 6.
> > > >
> > > >
> > > > > However, besides the projection method, we proposed another implementation variation of Hodge1Lap (namely ABS), see line 345 of main text and line 360-362 of supplementary. [...]
> > > > > In the original text, this implementation can be described as $\sum_i MLP(|v_i|)$, where [...] and $v_i$ refers to the interested eigenvectors (e.g. orthogonal basis of kernel space of $L_1$, in case that we are discussing detecting cycles)
> > > >
> > > > I feel this design choice seems important to go beyond the limitation above but I don't think this design is clearly described in the original manuscript.
> > > > In particular, I am still unsure how the vector $v_i$ is built: the paper mentions "summing over absolute values of eigenvectors" (l. 345), do you mean  $v_i = \sum_j |u_j|$ where $[u_j]_j$ is the eigenvectors basis described in Theorem 3.9?
> > > > As argued above, the projection $v_i = P\_{proj,i}e$  of just the constant vector $e$ only leads to 0 or 1 features on the edges so it is not a good choice of $v_i$.
> > > >
> > > > Moreover, is the construction $\sum_j MLP(\sum_i a_{ij} |v_i|)$ described in the paper? I quickly checked the manuscript but I couldn't see it (sorry if I missed it). How do you apply this construction over different graphs? (Do I understand correctly that the index $i$ ranges over the unique eigenvalues? Then, the $L_1$ matrix of different graphs might have different number of unique eigenvalues, right?).
> > > >
> > > > > the elements corresponding to edges from the same cycles have identical absolute values, while those of edges from different cycles (almost surely) have different absolute values
> > > >
> > > > Could you clarify what the values will be?
> > > > If we use $v_i = \sum_j u_j$ where $[u_j]_j$ is the eigenvectors basis described in Theorem 3.9, the entries of $v_i$ encode the cycle length (according to Thrm 3.9). Why do you think these values are almost surely different? Cycles with similar lengths seem like something that can occur commonly in general graphs.
> > > >
> > > > Moreover, if you assume the basis in Theorem 3.9, how do you compute it in practice (since numerical methods will return a random orthogonal basis, not necessarily this one)? This seems related to the typical problem of finding a sparse basis for homology I was referring to in my initial review.
> > > >
> > > > Overall, I feel like some more space in the main paper could be dedicated to clarify these important design choices

---

> > > > > ### Author Response · Authors · 2023-08-19
> > > > >
> > > > > Thank you for your suggestions! We will definitely include these systematic discussion as well as details of Hodge1Lap construction (including random projection technique) in the revision paper.
> > > > >
> > > > > As for your questions, we think the followings can be answered together: (1) do you mean $|v_i|=\sum_j u_j$ where $u_j[j]$ is the eigenvectors basis described in Theorem 3.9? (2) Could you clarify what the values will be? Why do you think these values are almost surely different? (3) if you assume the basis in Theorem 3.9, how do you compute it in practice (since numerical methods will return a random orthogonal basis, not necessarily this one)?
> > > > >
> > > > > The answer would be: $v_i$ are **not** $u_j$ described in Theorem 3.9, instead, they are **arbitrary random basis**, i.e. linear combinations of $u_j$, as long as they span the same subspace as $u_j$ (they don't even have to be orthogonal). One example would just be the random orthogonal basis returned by numerical methods, obviously, both these random basis $v_i$ and those $u_j$ in Theorem 3.9 span the kernel space of $\mathbf L_1$. We want to **make use of the randomness** to make the values corresponding to different cycles **different** from each other. Let us see an example, suppose we have a graph with ten edges, where edge $0,1,2,3$ form a 4-cycle, edge $5, 6, 7, 8, 9$ form a 5-cycle, and edge $4$ connects these two cycles. The basis from Theorem 3.9 will be $u_1=[1, 1, 1, 1, 0, 0, 0, 0, 0, 0]^T, u_2=[0,0,0,0,0,1,1,1,1,1]^T$. However, the $v_i$ we use here are different from them. Instead, they are any random orthogonal basis (also linear combinations of $u_i$), for instance, $v_1=0.1 u_1+0.9u_2=[0.1,0.1,0.1,0.1,0,0.9,0.9,0.9,0.9,0.9], v_2=-0.4u_1+0.6u_2=[-0.4,-0.4,-0.4,-0.4,0,0.6,0.6,0.6,0.6,0.6]$. Then obviously in either $|v_1|,|v_2|,|v_1+v_2|$, the values of different cycles are different, so that $\sum_{i=1}^2 MLP(|v_i|)$ would result different values for different cycles, but the same for edges within one cycle. Therefore, the model is able to distinguish that there are two different cycles, as different cycles (almost surely) correspond to different values. Moreover, the random projection technique is applied to improve the generalization power of the ABS form of Hodge1Lap, i.e. by training with random coefficients $\alpha_{ij}\sim \mathcal N(0,1)$, the model do not rely on specific concrete values of $v_i$, but make use of statistical information about these cycles. In our experiments, we observe that either the deterministic version (directly use the basis returned by numerical values) or the random projection version works well. We report the former implementation as "ABS" Hodge1Lap in Table 1 of main text.
> > > > >
> > > > > Again, we admit that the above "ABS" implementation is not basis invariant (i.e. the output is determined by input basis $v_i$), while the projection method described in the main text is basis invariant (as we project the vectors into the kernel space spanned by the basis). However, the "ABS" version can distinguish different cycles, and may have more generalization power due to randomness. Experiments show that both two designs have good performance. Additionally, it's remarkable that traditional LapPE that make use of eigenvectors of graph Laplacian (Hodge-0 Laplacian) suffer from the same problem (the output relies on the choice of basis), but LapPE is still widely adopted, e.g. in GPS and many other experiments. In this sense, our "ABS" version of Hodge1Lap is the natural extension of traditional LapPE.
> > > > >
> > > > > We will definitely include these detailed discussion into the new version of our paper to let readers better understand. Sorry for the confusion and hope that we have addressed your questions.

---

> > > > > > ### Comment · Reviewer_r8fz · 2023-08-20
> > > > > >
> > > > > > Thanks for the clarification!
> > > > > >
> > > > > > Again, I encourage the authors to include this discussion in the main paper, since it seems particularly important.

---

### Official Review · Reviewer_1k26 · 2023-07-09

**Soundness:** 3 good
**Presentation:** 2 fair
**Contribution:** 2 fair
**Rating:** 5
**Confidence:** 2

**Summary:**

Positional and structural encodings have been widely studied, especially in the context of graph transformer architectures where such encodings often make up for the lack of inductive bias provided by the original graph, due to dense connectivity in the attention mechanism. RWSE (random walk structural encodings) are common approach based on random walk probabilities of a node returning to itself after differing numbers of steps. The authors build on this work, drawing from theory of simplicial complexes and Hodge deRham theory, to propose a set of edge-level positional encodings. They provide theoretical analysis of the proposed technique and evaluate its effectiveness by incorporating the positional encodings into MPNNs and/or graph transformers on a number of datasets, including ZINC, as well as datasets from the OGB and LRGB collections.

**Strengths:**

The authors consider an interesting question around positional and structural encodings, which have been widely studied. The authors introduce an interesting approach using Hodge Laplacians that build on previous work.

**Weaknesses:**

-The paper seems a bit rushed and has some typos. The paper can benefit from some careful proofreading.
-Given that this work proposes GNNs based on techniques related to simplicial complexes, I would have liked to see experimental comparisons to some existing architectures based on simplicial complexes (e.g., [3], [4], [28] in the paper) or other topological baselines.
-The experimentation on OGB benchmarks has very little explanation. I would like to see more details about the setup and design choices.

I find the authors' approach interesting but think the experimental evaluation section could be stronger in general. I'd be willing to raise my score upon answers to the below questions and some actions to address the points stated above.

**Questions:**

-The experiments on ZINC appear to use a standard MPNN (GINE) for the purposes of testing out the authors' random walk simplicial complex based approach. On the other hand, the experiments on OGB appear to be based on GPS framework which uses a combination of graph transformers and MPNNs. Is there a reason for this choice? Have you tried using the proposed random walk simplicial encodings with GPS/graph transformers on ZINC? This would be interesting, as positional and structural encodings are particularly critical for graph transformer architectures where the inductive bias provided by the original graph is lost due to dense connectivity.
-Suggestion: It should be noted that for several of the LRGB datasets you consider in the appendix, there are improved results, so the results by the authors are not quite SOTA. For instance, the recent Exphormer [1] paper improves on GraphGPS numbers for most of the LRGB datasets. Please include this baseline..
-It is not clear to me which results are new among the theorems (e.g., Thms. 5.2, 5.3 and the ones in the appendix) as opposed to previously known. It would be good for the authors to clarify this


[1] Exphormer: Sparse Transformers for Graphs - Shirzad, Velingker, Venkatachalam, Sutherland, Sinop. ICML 2023.

**Limitations:**

I could not find a discussion about the limitations of the authors' proposed technique. However, it is possible that I have missed it. Societal impacts are not relevant to this work.

---

> ### Author Rebuttal · Authors · 2023-08-09
>
> Thank you for your constructive review. We address them all in the following.
>
> Weaknesses:
>
> (1) Typos: We will carefully proofread the paper, fix the typos and improve the writing.
>
> (2) Comparisons to existing architectures based on simplicial complexes (e.g., [3], [4], [28]): First, [3, 4, 28] are network architectures while our methods are PE/SE, so they are in some sense orthogonal to ours and cannot be directly compared. Nevertheless, [3] is actually just CIN, which is already compared in the tables of our paper. CIN is SOTA on ogbg-molhiv, but our methods generally perform better than CIN, for example on Zinc. Note that CIN is also heavily hand-crafted as it extracts cellular complexes, while our methods are more general and do not rely on prior/domain knowledge. [4] and [28] mainly conduct experiments on simplex datasets, which cannot be easily generalized to graph datasets; in comparison, our methods can be easily generalized to simplicial complexes. We test on synthetic dataset from [4], and show that MPSN(tanh) augmented with Hodge1Lap can achieve over $99.5\pm 0.3$% test accuracy, exceeding SOTA baseline ($95.2\pm1.8$%) in Table 1 of [4] (MPSN). This is consistent with the experimental analysis in [4], that the model needs to learn two approximately orthogonal components in kernel space of $\mathbf L_1$, while Hodge1Lap can perfectly achieve this target (however not by projection as described in our main text, but learn two eigenvectors jointly through a linear layer; the invariance and equivariance are guaranteed by MPSN base model).
>
> (3) Explanation on OGB experimentation: The setup of experiments on OGB are the same as GPS for a fair comparison. However, the performance on ogbg-molhiv and ogbg-molpcba are less convincing to evaluate models: SOTA results on ogbg-molhiv typically involve manually crafted structures, including molecular fingerprints, GSN and CIN, while general methods and expressive models usually suffer from overfitting and cannot generalize well on the test set. We apply our Hodge1Lap and EdgeRWSE to both GatedGCN and GPS (consisting of GatedGCN and Transformer) and show that our methods can improve both architectures. However, the overfitting phenomenon still exists.
>
> Questions:
>
> (1) Why choosing GINE (MPNN) as backbone: On Zinc dataset, GINE and GPS (with transformer) have similar performance according to the original GPS paper [23]. Therefore, to reveal that even a simple MPNN-like architecture can benefit from our proposed PE/SE, we choose GINE as the backbone model on Zinc. It's true that PEs are indeed more beneficial for graph transformers. In our new experiments (see Table 1 in the new pdf), we apply our methods to: (a) GAT; (b) PNA; (c) SSWL+; (d) GPS; (e) GRIT, whereas the latter two are recent state-of-the-art graph transformers. Our new experiments show that our methods are beneficial for a wide variety of architectures, including message-passing based and attention based models. By integrating Hodge1Lap, the test MAE on zinc decreases from $0.070$ to $0.064$ for GPS, from $0.059$ to $0.057$ for GRIT (which is SOTA performance on Zinc to the best of our knowledge). Similar improvements are also observed for EdgeRWSE, which verifies that our PE/SE are benificial for graph transformers. Remarkably, GPS and GRIT benefit from our edge-level PE in different ways. GPS adopts a standard transformer, so the only way to integrate edge features is through the local message-passing module (GINE in our experiments). In comparison, GRIT explicitly reserve the expanded edge features for every possible node pairs, which is able to process edge features without a local module. As for OGB benchmarks, we choose GPS as the backbone because it generally performs better than MPNN families.
>
> (2) Including the recent Exphormer: We have included this baseline. The performance of Exphormer with/without Hodge1Lap and EdgeRWSE are in Table 2 of the new pdf. However, we re-run the Exphormer experiments and find that the actual performance we obtained are weaker than the reported ones in their paper. We are still working on this point.
>
> (3) Which theoretical results are new: Section 1 and Section 2 in the appendix give the background information and summaries of previous results. We formally present our new conclusions and proof in **Section 4-6 of the main text** and **Section 3 of the appendix**. To the best of our knowledge, all these formally presented theorems in the main text (Theorem 4.1, 5.1 - 5.3) and Section 3 of the appendix (Theorem 3.1 - Theorem 3.13) are **newly proposed**. We make close connections of these mathematical properties with the Weisfeiler-Lehman expressive power hierarchy, and we also provide rigorous proofs for all these new theorems. Nevertheless, there may be cases where part of conclusions or proofs are borrowed from existing results, which have all been annotated in the statement or proofs in our original text.

---

> > ### Author Response · Authors · 2023-08-21
> > **Looking forward to your reply**
> >
> > Dear reviewer,
> >
> > We sincerely appreciate your review points. However, while other reviewers have positively acknowledged our efforts, you haven't replied to our rebuttal yet. We put great effort into your review points and tried our best to address your concerns during the rebuttal period. Therefore, we genuinely hope that you could reply to us and give your final decision, especially given that you have promised to raise your score if your concerns are addressed. Your response is of great importance for our paper to be evaluated fairly.
> >
> > Thank you!

---

> > > ### Comment · Reviewer_1k26 · 2023-08-22
> > > **Acknowledgment**
> > >
> > > Thank you for providing a detailed response. You have substantially addressed my concerns, and I am increasing my score accordingly.

---

> > > > ### Author Response · Authors · 2023-08-22
> > > > **Thank you**
> > > >
> > > > Thank you for acknowledging our rebuttal. We sincerely appreciate your review points and acknowledgment, which definitely improve our paper.

---

> ### Author Response · Authors · 2023-08-19
> **We are looking forward to your reply**
>
> We thank reviewer for the review points, and we have answered your questions in detail while making every effort to improve the experiments. We are looking forward to your reply to see if we have properly addressed your concerns, as you said you are wiling to raise your score if these questions are addressed. Your response will definitely help to make a more reasonable decision on our paper.

---

### Official Review · Reviewer_Djvd · 2023-07-24

**Soundness:** 3 good
**Presentation:** 2 fair
**Contribution:** 3 good
**Rating:** 6
**Confidence:** 3

**Summary:**

Authors propose leveraging the random walk on different orders of the simplicial complex domain to enhance graph neural networks both in their performance and in the understanding of their theoretical expressivity.

$\textbf{Node level}$

Authors frame existing positional encoding (PE) and structure encoding (SE) methods in the context of random walk on 0-th order simplices and the traditional graph Hodge Laplacian $\mathbf{L}_0$. Using this framework, they provide a theoretical expressivity bound for RWSE, a SE method, and provide an intuitive explanation of its nature. They summarize PE methods and their expressivity bounds from the literature.

$\textbf{Edge level}$

Authors introduce two novel edge-level PE and SE methods:
- EdgeRWSE, which directly generalizes random walk on nodes to edges, with 3 variations: full (upper and lower adjacency), down adjacency directed, and down adjacency undirected. Authors test the directed/undirected down variations on graph regression (not competitive). Authors test a variation on graph-level classification (not competitive, overfits).
- Hodge1Lap, the first sign / basis invariant edge-level PE, with 1 variation based on the spectra of the Hodge 1 Laplacian. Authors test sign invariant and sign / basis invariant Hodge1Lap on graph regression (SOTA). Authors test a variation on graph-level classification (competitive).
- Random Walk Message Passing, a new random walk method based on a metric of node feature similarity. Authors test it on graph regression (SOTA) and show that it can improve the performance of Hodge1Lap (competitive).

$\textbf{k-level}$

Authors extend the theoretical framework of the random walk on simplicial complex arbitrary order $k$ by proposing:
- a method that extends RWSE to $k$-order.
- inter-order random walk, a method that utilizes the upper/lower boundary neighborhoods of a $k$-simplex.

Supplementary materials provide mathematical proofs for expressive bounds, further theoretical analysis of proposed methods, explain RWMP and provide further experiments on MNIST, CIFAR10, and LRGB.

**Strengths:**

- This paper presents a comprehensive and wholesome approach to higher-order random walk, addressing 0, 1, and higher orders in a systematic manner.
- It offers a thorough and detailed background in preliminaries, providing a solid reference for the underlying concepts. The supplementary materials are additionally thorough.
- Authors provide intuitive explanations for the theoretical bounds and the functioning of the PE/SE methods. Reframing pre-existing methods in a fresh, accessible way is a valuable contribution. In the same vein, observed performance increases are supported by a sound theoretical justification, lending credibility to the results obtained.
- The regression experiments conducted on top of the GINE model are convincing, demonstrating the effectiveness of the proposed methods in graph regression tasks.
- The introduction of the clever new method RWMP opens the door to a new and flexible class of random walks based on the concept of a "feature distance" metric. This could be the first such method of many, as new metrics are defined on different feature orders with different probabilities.

**Weaknesses:**

- Implementing the (appropriate) proposed methods on higher-order datasets would have been a significant contribution. Based on the increased performance of Hodge1Lap which benefits from upper adjacency schemes, I would have been very interested to see the full EdgeRWSE, and the inter-order random walk in action. Lifted benchmark datasets, including Zinc 12k [1] are common in topological deep learning research to allow for comprehensive testing of proposed methods.
- While the paper introduces a valuable theoretical bound for a pre-existing method at the node level, this remains a case-by-case study of theoretical expressivity. This section would have been much more meaningful if it generalized this special case to a systematic method for bounding PE/SE methods from the random walk framework.
- Authors make no mention of other discrete topological domains, cellular/cubical complexes being the most obvious, and how their proposed methods extend to datasets defined on these domains. This low-hanging fruit would be an important contribution to the cellular complex domain which only differs from the simplicial complex in the construction of higher-order features.
- It would have been preferable to observe the proposed methods integrated with a wider range of neural network architectures, beyond just GINE, GPS, and GatedGCN. While the authors argue that the simplicity of these architectures showcases the power of their methods, I am still curious to see how these techniques impact the performance of more complex models like transformers and subgraph variations. Are there still significant improvements observed in these cases?
- There are many syntax and grammatical errors in the text that could be easily caught with spell-checking software like Grammarly. Here are $\textit{some}$ examples :
    - line 106: "there are still limited research" should be "there is still"
    - line 108: "establish several connection" should be "establish several connections"
    - line 109: "provide theoretically expressive bound" should be "provide a theoretically expressive bound"
    - line 129: "resistance in electrical network" should be "resistance in an electrical network"
    - line 484 in supplementary materials: "an function" should be "a function"

1. Bodnar, Cristian, et al. "Weisfeiler and Lehman Go Cellular: Cw networks." Advances in Neural Information Processing Systems 34 (2021): 2625-2640.

**Questions:**

- Could the proposed methods generalize to discrete domains without orientation, where the adjacency matrix is set-type (all positive) as in hypergraphs? And if so, how would their design/performance compare to previously introduced methods [1,2]?
- Is there an intuitive explanation for the state-of-the-art (SOTA) results achieved in the long-range graph benchmark task, particularly in relation to the additional reach that random walk methods gain from neighborhood structures? From my perspective, these impressive results should have been included in the main text. It could be direct proof of the benefits of simplicial complex-based random walks.
- What variations of EdgeRWSE and Hodge1Lap were used in the graph classification benchmark task? They are not specified in the table or in the text.
1. Carletti, Timoteo, et al. "Random walks on hypergraphs." Physical Review E 101.2 (2020): 022308.
2. Chitra, Uthsav, and Benjamin Raphael. "Random walks on hypergraphs with edge-dependent vertex weights." International conference on machine learning. PMLR, 2019.

**Limitations:**

- The authors highlight the overfitting behavior of EdgeRWSE in graph classification. It would have been interesting to see how different variations of EdgeRWSE compare in terms of overfitting when applied to datasets with higher-order features. Additionally, it would have been valuable to understand if this overfitting limitation is also at play in the non-competitive performance on Zinc 12k.
- A significant limitation left unaddressed in the paper is the single topological domain. While the paper focuses on simplicial complexes, all of its proposed methods and Hodge Laplacian theory could fit into a larger framework of topological domains, including cellular complexes and hypergraphs. Each random walk could be generalized to such domains and tested on respective datasets. If any theoretical reasons prevent such an extension, the authors should clearly outline and explain the reasoning behind this limitation.

---

> ### Author Rebuttal · Authors · 2023-08-10
>
> Thank you for your valuable review and constructive points. We address them all as follows.
>
> Weakness
>
> 1. Implementing higher-order and inter-order random walk: Thank you for your advice. We have newly implemented full EdgeRWSE and inter-order random walk in experiments, which we believe is huge contribution. (1) **Full-EdgeRWSE:** we implement Full-EdgeRWSE and show that a GINE augmented with it achieves $100\%$ performance on synthetic datasets EXP and SR25. To fully distinguish EXP and SR25, the model need to be more expressive than $1$-WL and $2$-FWL, respectively. This show that Full-EdgeRWSE is (partly) more powerful than $2$-FWL ($3$-WL), which is consistent with our theory. In comparison, 1-down EdgeRWSE that does not consider 2-simplices fail ($0\%$) to distinguish any strongly regular graphs in SR25. The performance of Full-EdgeRWSE on zinc dataset is similar to 1-down EdgeRWSE with a GINE base model. This is because 2-simplices (or triangles) are actually rare in molecules, so Full-EdgeRWSE makes no significant difference than 1-down EdgeRWSE. (2) **Inter-order random walk:** we implement the inter-order random walk among $0,1,2$-simplices. The random walk probability matrix is the normalized $\mathcal A_2$ in our main text. We compute $\mathcal A_2,\dots, \mathcal A_2^T$ ($T=20$ for Zinc) and embed the diagonal elements for both node and edge features. The performance of GINE augmented with Inter-RWSE is significantly better than pure GINE, but is slightly weaker than simultaneously applying NodeRWSE and EdgeRWSE (which are separately computed without inter-order communications). We attribute it to: (i) lack of tuning hyper-parameters such as $T$, and (ii) the return probabilities are much less in inter-order random walk, as there are totally $n+m$ possible targets, instead of $n$ for 0-random walk and $m$ for 1-random walk. This makes the model harder to learn meaningful structure information and may require larger $T$.
>
> 2. Case by case analysis on node-level random walk: We provide the following conclusions for node-level PE/SEs:
>  - RWSE is strictly less powerful than $2$-FWL.
>  - SPDPE (shortest path distance) is strictly less powerful than $2$-FWL.
>  - RDPE (resistance distance) is strictly less powerful than $2$-FWL, but more powerful than SPDPE in distinguishing non-isomorphic distance-regular graphs.
>  - RRWP is strictly less powerful than $2$-FWL, but is strictly more powerful than SPDPE.
>  - HK (heat kernel) is strictly less powerful than $2$-FWL.
>
> The second and third conclusions are from [30]. RRWP is originally proposed in [5], and we newly bound it with $2$-FWL. We will provide full proof in the revision paper. The core idea of the proof is that 2-FWL can simulate the multiplication and injective
> transformations of matrix (including adjacency matrix $\mathbf A$), so that $2$-FWL is able to obtain all the information of these PE/SE (e.g. degree, triangles, 4-cycles). Although node-level random walk is not our main focus, we believe the above case-by-case analysis is valuable to understand the limitations of PE/SE based on node-level random walk.
>
> [30] Rethinking the expressive power of gnns via graph biconnectivity.
>
> [5] Graph Inductive Biases in Transformers without Message Passing.
>
> 3. Other discrete topological structures: In line 399-402 of supplementary, we do mention other discrete topological structure such as cellular complex. We do not generalize our methods to these domain in the original version because of the limited space, and that our original contents are already fairly rich. It's also harder to theoretically analyze the expressive power of all possible cellular complexes case by case, since the definition of these topological structures are rather flexible. However, we do implement (the first-time ever) SE based on random walk on cellular complexes in our new experiments. As simple version, we treat edges as $1$-cells and extract all the rings as $2$-cells. In this CellularRWSE, one is able to transit from one source edge to (i) another target edge sharing one node, or (ii) another target edge that is in the same $2$-cell as the source edge. The performance is slightly better than EdgeRWSE (see Table 1), which may due to the capability to distinguish higher-order structures (rings).
>
> 4. Applying methods to more architectures: In our new experiments, we apply our PE to: (a) GAT; (b) PNA; (c) SSWL+; (d) GPS; (e) GRIT, where SSWL+[3] is one subgraph GNN backbone and the latter two are graph transformers. We see significant improvements for all these kinds of base models, please refer to Table 1 and common comment for more details.
>
> [3] A Complete Expressiveness Hierarchy for Subgraph GNNs via Subgraph Weisfeiler-Lehman Tests.
>
> 5. Sorry for the grammar mistakes, which we will correct in the final version.
>
> Questions
>
> 1. Generalization: Our methods indeed can be extended to discrete domains without orientations, which we also pointed out in our paper. In detail, for HodgeLap families, the Hodge Laplacians can be calculated whenever incidence matrix $\mathbf B_k$ (or boundary operator) is defined, see equation (5-6) in supplementary. For spatial domain random walk families, we can perform undirected random walk on these structures without orientation (e.g. EdgeRWSE for undirected graph, as described in line 169-173 in main text). They can all be naturally applied to hyper-graphs.
>
> 2. Long range information: Random walk methods are actually capable of capturing long-range information. For examples, a K-step node-level random walk is able to explore the K-hop subgraph around the root node. In our experiments, the number of steps are typically set as K=20, so it is able to capture long-range information and perform well on LRGB datasets.
>
> 3. Variations: We primarily explain the variations of Hodge1Lap and EdgeRWSE in Section 5.3 (line 558-569) in supplementary. If not specific, we use undirected version of EdgeRWSE, and projection implementation of Hodge1Lap.

---

> > ### Comment · Reviewer_Djvd · 2023-08-15
> >
> > Thank you for your answer and the new experiments.
> >  Regarding 2., section 4 of the paper would benefit from this concise list of expressivity theorems. Regarding 3, this gained performance form cellular representation is so exciting! Cellular complexes do in fact have a strict definition as a topological domain, explained well in a paper already cited [3. Weisfeiler and Lehman Go Cellular:]
> > Regarding question 1, I think it would be worth explicitly mentioning that this method naturally extends to hypergraphs. It is not a trivial question and would widen this work's contribution to the field without requiring any additional proofs or experiments.
> > Regarding question 3, the paper should absolutely be specifying what versions of EdgeRWSE and Hodge1Lap it refers to in all experiments, without assuming the reader can figure it out themselves. Naming the method used is a critical component to any Result section.

---

> > > ### Author Response · Authors · 2023-08-16
> > >
> > > Thank you for your valuable review points and suggestions! We have made every effort to address all your questions and implement all your suggestions. We are glad at your acknowledgement. Regarding 2, we will add this list of expressivity theorems of node-level PE to section 4 of our paper revision. Regarding 3, we will include the discussion and results on cellular complex extensions and give some preliminary expressivity results in the revision. Regarding question 1, we appreciate your suggestion and will explicitly explain in the revision that our methods naturally extend to hypergraphs. Regarding question 3, we will explain the variations of our methods in the paper.
> > >
> > > Hope that we have addressed all your concerns. Please inform us if you have more questions.

---

> > > ### Author Response · Authors · 2023-08-19
> > > **Thank you**
> > >
> > > We sincerely appreciate the valuable suggestions by reviewer. We will include all these new theoretical results, experiments and discussions during rebuttal to make our paper more contributive. Now that since we have solved all your questions and implemented all the suggestions you have put forward (there are even more results in our replies to other reviewers), we are wondering if you are willing to raise your score. Thank you!

---

> > > > ### Comment · Reviewer_Djvd · 2023-08-21
> > > >
> > > > I thank the authors for largely implementing my feedback. I have modified my score accordingly.

---

> > > > > ### Author Response · Authors · 2023-08-21
> > > > > **Thank you for acknowledgement**
> > > > >
> > > > > Thank you for your valuable acknowledgement! We sincerely appreciate your feedbacks, and we believe that the final version of our paper will be more contributive after we include these new results.

---

### Author Rebuttal · Authors · 2023-08-09

We sincerely thank all reviewers for their insightful and valuable review points. We add extensive experiments and report the results in the **pdf** attached to this common response. We answer some common questions in this response. Due to limited space, some experimental results and answers to the reviewers' questions are in individual replies.

**More backbone models**

In our new experiments (see **Table 1**), we apply our methods to: (a) GAT [1]; (b) PNA [2]; (c) SSWL+ [3]; (d) GPS [4]; (e) GRIT [5], where GAT and PNA are local (message-passing based) models, SSWL+ is a subgraph GNN, and the latter two are recent state-of-the-art graph transformers. We report the results over 5 different seeds for every setting. For both GAT and PNA that are relatively weak models, our Hodge1Lap and EdgeRWSE both decrease test MAE by around $50\%-70\%$. For the stronger backbone models SSWL+, GPS and GRIT, integrating Hodge1Lap and EdgeRWSE still improve their performances, resulting in highly competitive and even **new SOTA** results. For example, GRIT with Hodge1Lap achieves a **record-breaking** 0.057 MAE on Zinc-12k. All these results make it convincing that our Hodge1Lap and EdgeRWSE are capable of improving an extremely wide range of GNN models.

**Synthetic datasets**

To verify the theoretical expressive power of Hodge1Lap and EdgeRWSE, we conduct experiments on two synthetic datasets: EXP and SR25. EXP contains 600 pairs of non-isomorphic graphs that 1-WL and 2-WL fail to distinguish. SR25 contains 15 non-isomorphic strongly regular graphs (i.e., 105 non-isomorphic pairs) that 3-WL fails to distinguish. An accuracy of $50\%$ on EXP and $6.67\%$ on SR25 suggest the model fails to distinguish any non-isomorphic graphs in the dataset. From **Table 3** of the attached pdf, the $1$-WL equivalent GINE backend model can fully distinguish EXP and SR25 when augmented by Hodge1Lap and Full-EdgeRWSE, indicating that both methods can distinguish non-isomorphic graph pairs that $3$-WL (and $2$-FWL) fails, which is consistent with our theoretical analysis.

**Explanation on implementation details**
 - For SSWL+, we add edge PE/SE features embedded by a unique linear projection to the original edge features in every layer.
 - For GRIT, the PE is embedded by a linear layer and then directly added to the edge features. For virtual edges, the PE values are padded as zeros.
 - Our methods are applicable to higher-order GNNs, including simplicial networks and 2-$\delta$-FWL equivalent classes, while others (SPD, RD) cannot. Our methods are also more sparse.
 - Our methods can be applied to any base model without increasing much computation cost, and can be applied together with other types of PE/SE.

**Higher-order and inter-order implementation**

To make our theories for higher-order and inter-order random walk practical and more contributive, we implement Full-EdgeRWSE and $0-2$ inter-order random walk. Besides, we also generalize our random walk based theories to cellular complexes and implement Cellular-RWSE. We verify these new methods with the GINE backbone on the Zinc dataset. The results are presented in the **first block of Table 1** in the attached pdf. The implementation details are in our response to reviewer Djvd. The competitive results demonstrate the effectiveness of these new methods.

We make every effort during the rebuttal period to implement the new methods and supplement these new experimental results according to the reviewers' suggestions. We hope they can address your concerns and further strengthen our work.

[1] Graph attention networks.

[2] Principal neighbourhood aggregation for graph nets.

[3] A Complete Expressiveness Hierarchy for Subgraph GNNs via Subgraph Weisfeiler-Lehman Tests

[4] Recipe for a general, powerful, scalable graph transformer.

[5] Graph Inductive Biases in Transformers without Message Passing.

[6] Exphormer: Sparse Transformers for Graphs - Shirzad, Velingker, Venkatachalam, Sutherland, Sinop. ICML 2023.

---

### Decision · Program_Chairs · 2023-09-21

**Decision:**

Accept (poster)

**Comment:**

This paper proposes random walks on different orders of simplicial complexes to enhance the expressive power of graph neural networks. The reviewers appreciated the novel idea, the nice presentation, and the theoretical analysis. Upon reading the paper myself, I also think the empirical results are quite strong for such a simple idea. I recommend acceptance for this work.